

# Environments and lifting mechanisms of cold-frontal convective cells during the warm-season in Germany

George Pacey[1], Stephan Pfahl[1], and Lisa Schielicke[2]

[1]Institute of Meteorology, Freie Unversität Berlin, Berlin, Germany
[2]The University of Western Ontario, Department of Physics and Astronomy, London, Canada

**Correspondence:** George Pacey (george.pacey@fu-berlin.de)

**Abstract.** Convection often initiates in proximity to cold fronts during the warm-season, but how various processes favour convective initiation at different regions relative to the front is still not well-understood. By combining automatic front detection methods and a convective cell tracking and detection dataset, the environments and availability of different lifting mechanisms are analysed. Our results indicate that pre-surface-frontal cells form in the environments with the highest surface dew points and convective available potential instability (CAPE). At other front relative regions, cells form in environments with lower CAPE and surface dew points, though still significantly higher than regions without cells. Mid-level relative humidity discriminates particularly well between post-frontal cell locations and regions without cells. Pre-surface-frontal cells form in environments with the strongest large-scale lifting at 850 hPa and 700 hPa and also with the strongest convective inhibition. We also observe importance of large-scale lifting post-frontal, particularly at 500 hPa. Observational sunshine duration data indicate less sunshine before cell initiation compared to regions without cells at most front relative regions, which highlights that solar heating may not be responsible for the majority of cold-frontal cell initiation. The results in this study are an important step towards a deeper understanding of the drivers of cold-frontal convection at different regions relative to the front.

## 1 Introduction

What exactly drives convective cells to develop in time and space is not currently well-understood. There has been improvement in recent times of the representation of convective initiation in numerical models; primarily due to convective permitting models (CPMs) at increased resolution. However, biases regarding the positioning, timing and intensity of convection still remain (e.g. Kain et al., 2008; Klasa et al., 2018). What is clear is the necessity of three primary ingredients to facilitate deep moist convection: moisture, lift and instability (Doswell et al., 1996). Vertical wind shear is also required to allow convective storm organisation (Markowski and Richardson, 2010). Moisture, instability and wind shear can be directly measured in the storm's environment using vertical profiles. Traditionally, this was done using observational proximity soundings (e.g. Brooks et al., 1994; Púčik et al., 2015; Kolendowicz et al., 2017; Taszarek et al., 2017). However, the advent of ERA5 reanalysis data on a global scale with a spatial and temporal resolution of 0.25 degrees and 1 hour, respectively, has enabled more localised studies where observational soundings are lacking (e.g. Taszarek et al., 2020; Calvo-Sancho et al., 2022: Poręba et al., 2022; León-Cruz et al., 2023). The precursor ERA-Interim also allowed analysis of convective storm environments (e.g. Taszarek



et al., 2018), albeit at a coarser spatial and time resolution.

Most previous studies have sought to better understand the environments in which convective storms form by analysing quantities such as mid- and low-level moisture, instability, wind shear etc, but few studies have analysed the availability of individual lifting mechanisms. Determining whether air parcels will be lifted to their level of free convection (LFC) is essential
for convective initiation and is often difficult to forecast. It is worth mentioning that moist air parcels reaching their LFC is not a guarantee of deep moist convection due to the possibility of entrainment (Morrison et al., 2022). Locating where and when convective cells initiate and the responsible mechanisms is essential for accurate prediction of convective hazards such as hail, strong winds and tornadoes. This is because if parcels are not lifted to their LFC and thus no CAPE is released then there is no convective cell and no possibility of severe convective storms (SCSs). This is regardless of whether there is a seemingly
favourable environment for SCSs (e.g. high CAPE and high vertical wind shear). Indeed, warm season forecast failures for severe convective storms are often related to difficulties anticipating convective initiation (Markowski and Richardson, 2010). Despite this, the environments in which SCSs form have arguably attracted more attention in previous literature compared to convective initiation.

Individual lifting (triggering) mechanisms such as sea-breezes, outflow boundaries and quasi-geostrophic (QG) forcing for ascent cannot be directly sampled in environmental vertical profiles. QG forcing for ascent is thought to be particularly relevant in proximity to cold fronts. Due to the weaker horizontal temperature gradients, frontal lifting is typically weaker during the warm-season. The literature would benefit from studies quantifying the relevance of frontal lifting at different regions relative to the front, especially during the warm-season. Previous literature has alluded to the role of solar heating combined with frontal
lift in ultimately determining where and when convection occurs in proximity to cold fronts (e.g. Doswell, 2001). Behind the front, where there is generally large-scale descending motion, solar heating seems to carry more importance (e.g. Weusthoff and Hauf, 2008; Pacey et al., 2023). However, when convection occurs the large-scale descent may be weaker than climatology or more localised areas of ascending motion may aid the development of convective cells.

In this study, individual lifting mechanisms such as solar heating and quasi-geostrophic forcing for ascent as well as the environments of cold-frontal convective cells are analysed. Vertical wind shear is also analysed as it may both positively or negatively affect convective initiation (Peters et al., 2022). Moist updrafts in sheared environments have been shown in simulations to have lower terminus heights than their non-sheared counterparts (e.g. Peters et al., 2019). On the other hand, updrafts are typically wider in sheared environments so may be less susceptible to entrainment. Here, a special focus is placed on the
variation of the environment and lifting mechanisms depending on the region relative to the front. Pacey et al. (2023) already showed large differences in the cell frequency and characteristics depending on the cell location relative to the front. For example, cells are between 4-5 times more frequent pre-surface-frontal compared to post-frontal. Furthermore, pre-surface-frontal cells are most likely to be associated with 55 dBZ convective cores and mesocyclones. We seek to better understand the differences observed in Pacey et al. (2023)'s climatology by delving into the environments and lifting mechanims of cold-frontal



convective cells. Rather than only focusing on convective cell environments (i.e. ERA5 grid points associated with convective cells), grid points not associated with convective cells are also considered. This allows analysis of how well certain variables can distinguish between grid points where convective cells *did* and *did not* develop. The separation into categories will be explained in section 2.3. The primary research questions addressed in this study are as follows:

Q1) How do convective cell environments vary across the front?

Q2) What is the relevance of quasi-geostrophic forcing for ascent (descent) and solar heating at different regions relative to the front?

Q3) By analysing the environments and lifting mechanisms can we explain the cell frequency at different front relative regions?

This paper is organised as follows:

Section 2 introduces the datasets and methods used in this study. Section 3 performs a statistical comparison of the environments and lifting mechanisms of convective cells depending on the region relative to the front. Section 4 puts the results in the context of the cell frequency at different front relative regions (as shown in Pacey et al., 2023).

## 2 Data and Methodology

A convective cell detection and tracking dataset (KONRAD; Wapler and James, 2015) and ERA5 data (Hersbach et al., 2018a, 2020) are combined between 2007–2016 for the months April–September in the German radar domain. An automatic front detection algorithm is applied to ERA5 (section 2.1) to build a time series of cold fronts. Cell environments and lifting mechanisms are analysed mostly using ERA5 data but also using sunshine duration station data (section 2.4.3). To understand the 80 differences in the environment and lifting mechanisms at different regions relative to the front, the distance in kilometres between the front and ERA5 grid points is derived. Grid points on the warm and cold side of the front are assigned a positive and negative distance, respectively.

### 2.1 Front Detection

Since the same methodology is used to detect fronts as described in Pacey et al. (2023) (their section 2.1), the methods are only 85 briefly reintroduced here. The front detection method is based on the Thermal Front Paramter (TFP) equation. The equation was introduced by Renard and Clarke (1965).

$$\text{TFP} = -\nabla|\nabla\tau| \cdot \frac{\nabla\tau}{|\nabla\tau|} \tag{1}$$

where $\tau$ is a thermodynamic variable (e.g., potential temperature or equivalent potential temperature).





A projection of the horizontal wind ($\mathbf{v}$) onto the frontal line is enabled using Equation 2 (Hewson, 1998).

$$v_f = \mathbf{v} \cdot \frac{\nabla(\mathrm{TFP})}{|\nabla(\mathrm{TFP})|} \tag{2}$$

The term $v_f$ is the horizontal wind ($v$) projected in the direction of the TFP gradient. The criteria used to detect cold fronts in this study are summarised below:

$$|\nabla\theta_e| > 6\,\mathrm{K}(100\,\mathrm{km})^{-1} \tag{A}$$

$$v_f > 1\,\mathrm{m\,s}^{-1} \tag{B}$$

$$L > 1000\,\mathrm{km} \tag{C}$$

The equivalent potential temperature is denoted by $\theta_e$ and the along front length by $L$. The overlap of the $\theta_e$ gradient threshold (condition A) and velocity threshold (condition B) is the front contour. As in Pacey et al. (2023), only synoptic fronts (~1000 km) are of interest. Therefore, fronts with an along front length ($L$) less than 1000 km are not included in the analysis (condition C). The frontal line is identified at the maximum of the equivalent potential temperature gradient by applying the following condition:

$$\mathrm{TFP} = 0 \tag{D}$$

The latitude and longitude of where TFP=0 is determined using interpolation. The distance between each adjacent point is calculated and summed across the whole line to give the front length ($L$). The four aforementioned criteria are applied to smoothed $\theta_e$ and horizontal wind fields at 700 hPa in ERA5. A smoothing function is applied to the fields whereby the nearest four neighbours of a grid point are averaged. The process is repeated 30 times to remove any local-scale features. The fronts are detected on the 700 hPa pressure level to avoid interaction with orography, which is a common issue in central Europe (Jenkner et al., 2010; their section 4.4). Furthermore, the 700 hPa level is further above the turbulent boundary layer.

The thresholds (conditions A, B and C) are stricter than some previous studies. Looser thresholds may increase the number of erroneously detected fronts. On the hand, higher thresholds generally reduce the number of fronts in the dataset but limit the dataset to synoptic-scale cold fronts. Overall, a dataset with a lower front count and higher percentage of correctly detected fronts is prioritised. Cold fronts are detected in a subsection of the European domain ([40N–70N, 20W–20E]). Examples of the detected cold frontal lines in Germany are shown in Figure 1. A total of 4 timesteps are shown across different days and years.

### 2.1.1 Surface front relative to the 700 hPa front

Due to the rearward slope of cold fronts with height, using a single pressure level to detect fronts (700 hPa in this case) requires some special consideration. The 700 hPa level is approximately 3 km above the surface and cold fronts have a mean slope of ~1:100 (Bott, 2023). This means that the surface front would be on average 300 km horizontally displaced ahead of the 700 hPa front. This is also supported by the mean maximum climatological surface convergence in ERA5 data (Pacey et al., 2023;





their Figure 3). While the slope of the front and corresponding surface front location relative to the 700 hPa front are likely
to vary per case study, we proceed assuming the climatological location of the surface front (hereafter surface front) is 300
km ahead of the 700 hPa front and use it as a reference point for this study. Furthermore, we will also use the terminology
pre-700-frontal and post-700-frontal to indicate a cell is on the warm and cold side of the 700 hPa front, respectively.

## 2.2    KONRAD Convective Cell Detection and Tracking Algorithm

KONRAD (KONvektionsentwicklung in RADarprodukten, convection evolution in radar products) is a convective cell de-
tection and tracking algorithm originally applied to 2D radar data in the German radar domain (Wapler and James, 2015).
KONRAD is run operationally by the German Weather Service (DWD) with a spatial and time resolution of 1 km and 5 min-
utes respectively. A convective cell is defined as an area with 15 pixels or more exceeding 46 dBZ. As the spatial resolution of
KONRAD is 1 km, 1 pixel ~ 1 km$^2$. The cell centre as well as the maximum north, south, west and east extent of the cell are
provided. Further details are available in Pacey et al. (2023) (their section 2.2). Unlike Pacey et al. (2023), additional definitions
such as the number of pixels exceeding 55 dBZ, lightning strike count and mesocyclone intensity are not used in this study.
Here, the convective cell definition is solely based on the criteria below:

$$Reflectivity \geq 46 \text{ dBZ}$$
$$Cell\ Area \geq 15 \text{ km}^2$$

## 2.3    Defining non-cell regions, cell regions and cell grid points

Rather than solely focusing on where cells occurred (cell grid points), the surrounding regions are also assessed (cell regions),
as well as regions where no cells occurred (non-cell regions). This comprehensive approach allows us to understand how well
certain variables can distinguish between grid points where convective cells *did* and *did not* develop. First, cell grid points
must be defined. The KONRAD dataset (section 2.2) has a spatial and temporal resolution of 1 km and 5 minutes, respectively.
ERA5 has a 0.25 degree spatial resolution and a 1 hourly temporal resolution, respectively. Spatially, ERA5 grid points within
the maximum north, south, west and eastern extent of the cell area are labelled as convective cell grid points. Since some
cells have a lower area than the grid size the bounds are increased by 0.125 degrees (half a grid point) to ensure every cell
is associated to at least one grid point. Applying this approach ensures that the area where the convection is occurring is
labelled as a convective cell grid point. Temporally, cells are associated to the timestep before the first cell detection time.
For example, a cell first detected between 14:00–14:59UTC is assigned to the 14 UTC timestep. This is to avoid sampling
the post-convective environment, which is particularly important for thermodynamic variables such as convective available
potential energy (CAPE).

To define the regions, bins are first created at different front relative distances. Following the approach from Pacey et al.
(2023), we only consider grid points within 750 km of the 700 hPa front. Using 100 km intervals there are a total of 15 bins.
Each ERA5 grid point within 750 km of a 700 hPa front detected between 2007–2016 and April–September is associated to





one of these bins. If a convective cell grid point is present within one of the 100 km bins at the current timestep but *not* at the given grid point then this grid point is labelled as a *cell region*. If no convective cell grid point is present within the 100 km bin at at the current timestep then this grid point is labelled *non-cell region*. The number of grid points associated to each category at different distances from the front is shown in Table 2. The total number of grid points in each bin varies since pre-frontal regions occur more often than post-frontal in Germany. This can be explained by cold fronts reaching southern parts of Germany less frequently, which then allows a larger number of post-frontal grid points to be sampled. Grid points in each category are visualised in Figure 1 for 4 independent timesteps. In the first example (Figure 1a) most grid points are labelled non-cell regions (green). Cells were only detected in the bins 300–400 km and 400–500 km, therefore most grid points in this region are labelled cell regions (purple). Grid points where convection occurred (cell grid points) are in yellow. Only timesteps with cells occurring in proximity to the front are shown in Figure 1, however timesteps with no cells are also included in the analysis. In this case, all grid points would be assigned the non-cell region category (green).

## 2.4 Variables

The variables analysed in this study broadly follow the ingredients based methodology proposed by Doswell et al. (1996); moisture, lift and instability. Wind shear is also considered as it may also both positively or negatively affect convective initiation (e.g. Peters et al., 2022) and is related to the organisation of convective cells (e.g. Markowski and Richardson, 2010). Large-scale and convective precipitation are analysed to see how ERA5 represents precipitation amounts across the front. A full list of the variables analysed in this study are shown in Table 1.

### 2.4.1 Convective Inhibition Dataset

The Convective Inhibition (CIN) parameter available from the Climate Data Store assigns a missing value if CIN exceeds 1000 J kg$^{-1}$ or if there is no cloud-base present (Hersbach et al., 2018b). So that a CIN value is present for all grid points, CIN is obtained from an alternative data source (thundeR; Taszarek et al., 2023). CIN is derived in thundeR using ERA5 model level data. Three model parcel departure levels are considered: most unstable CIN (MUCIN), mixed-layer CIN (MLCIN) and surface-based CIN (SBCIN). The CIN parameters are calculated by integrating negative parcel buoyancy between the parcel initialisation height and the LFC. The most unstable parcel refers to the parcel with the highest equivalent potential temperature between the surface and 3 km above ground level (AGL). The mixed-layer parcel is calculated by averaging the potential temperature and mixing ratio between the surface and 500 metres AGL and initialising from surface. The surface based parcel is nearest parcel to the surface.

### 2.4.2 Q-vectors

The quasi-geostrophic forcing for ascending and descending motion can be measured using the Q-vector convergence derived from the quasi-geostrophic omega equation (Hoskins et al., 1978). Q-vectors are derived using the Python package MetPy (May et al., 2022), which derives Q-vectors in the following way (Equation 3).





$$Q_i = -\frac{R}{\sigma p}\left[\frac{\partial u_g}{\partial x}\frac{\partial T}{\partial x} + \frac{\partial v_g}{\partial x}\frac{\partial T}{\partial y}\right]$$
$$Q_j = -\frac{R}{\sigma p}\left[\frac{\partial u_g}{\partial y}\frac{\partial T}{\partial x} + \frac{\partial v_g}{\partial y}\frac{\partial T}{\partial y}\right]$$

(3)

Where:

$Q_i$ : x component of the Q-vector

$Q_j$ : y component of the Q-vector

$R$ : Gas constant for dry air

$\sigma$ : Static stability parameter

$p$ : Pressure

$u_g$ : u component of geostrophic wind

$v_g$ : v component of geostrophic wind

$T$ : air temperature

The $u$ and $v$ components of the geostrophic wind are derived from the geopotential height field. The geopotential height fields are first smoothed using a simple smoothing function whereby the nearest four neighbours of a grid point are averaged. The process is repeated 50 times. The air temperature field ($T$) is also smoothed 50 times. The smoothing is required to filter out local scale features to be left with the large-scale circulation. Smoothing values between 10 and 100 were tested and 50 was selected as it showed a realistic and smooth frontal circulation with Q-vector convergence ahead of the front and divergence behind the front.

### 2.4.3 DWD Sunshine Hours Station Data

In section 3.6 solar heating is analysed using total incoming solar radiation from ERA5. Observational sunshine duration data from the German Weather Service (DWD) is also analysed to see if the two datasets are in agreement. The data was downloaded from the Open Data Server (Kaspar et al., 2019). The 10-minute station observations of sunshine duration (Deutscher Wetterdienst, 2024) are used for the years 2010–2016 between 09-18 UTC. Only ERA5 grid points within 15 km of a DWD station are considered as solar radiation can vary on small spatial scales.

## 3 Results

The environments and lifting mechanisms of cold-frontal cells are analysed at different regions relative to the front. A comparison is also made to cell regions and non-cell regions (see definitions in section 2.3). The mean of each variable is taken across each 100 km front relative bin for each category (i.e, cell grid points, cell regions and non-cell regions; Table 2).



## 3.1 Moisture

Surface dewpoints (2-metres above ground level in this case) are a measure of moisture availability at the surface. While sufficient moisture directly at the surface is not essential for convective initiation since convection can be elevated (e.g. Corfidi et al., 2008), surface dewpoints are a commonly used tool by forecasters. Lower dewpoint air generally requires more lifting to reach the lifting condensation level (LCL). Figure 2 shows that pre-700-frontal cells develop in environments with a surface dewpoint of around 15–16 °C on average. Cells 650–750 km ahead of the 700 hPa front, which is the furthest distance from the front considered, have the highest mean dewpoints. Post-700-frontal cells formed in environments with lower surface dewpoints ranging between 12-14 °C. Dewpoints at cell grid points are around 1 °C higher than at cell region grid points on average. The difference in the mean between cell grid points and non-cell region grid points is significant at all distances relative to the front. The difference between the cell grid point mean and non-cell regions mean ranges between 3-4 °C.

Mid-level moisture is also relevant for the initiation of deep moist convection since entrainment of dry environmental air can lead to updraft dilution, hence influencing updraft buoyancy. Recent work has shown that how susceptible the updraft is to dry environmental air is dependant on the updraft width below the LFC (Morrison et al., 2022). However, the updraft, which occurs on the storm-scale, will not be looked at in this study. The relative humidity (RH) is analysed at 700 hPa (hereafter RH700) as parcels have typically already passed the LFC at this level so entrainment is important in determining whether updrafts can reach upper levels of the atmosphere.

Cells form in environments with a mean RH700 of between 60–70% post-700-frontal and 70-85% pre-700-frontal (Figure 3). Cells have the largest RH between 50-250 km ahead of the 700 hPa front, which is also the case for cell regions and non-cell regions. The RH700 at cell grid points is significantly higher than non-cell regions at all distances from the front except in the region 50–150 km ahead of the 700 hPa front. A larger difference exists between cell grid points and non-cell region grid points post-700-frontal (up to 30% difference) compared to pre-700-frontal (up to 10% difference). This indicates that a larger enhancement of RH700 humidity is required to facilitate post-700-frontal cell development. On the other hand, the warm-sector typically already has high upper-level moisture content. Excluding the region 50-250 km, cell regions have slightly lower RH than cell grid points but are above non-cell region grid points. These results are consistent with environment studies of lightning in Europe which found that lightning is less favourable when mid-level relative humidity is low (Westermayer et al., 2017) and that 75% of lightning cases in Poland had RH700 of 65% or higher (Poręba et al., 2022; their Figure 11). The average RH between 850-500hPa (Figure 3) shows a very similar result as RH700.

## 3.2 Instability

CAPE is a measure of atmospheric instability; a prerequisite for deep moist convection (Doswell et al., 1996). It is also a very commonly used parameter in severe convective storm forecasting. The CAPE used in this study is the CAPE from ERA5 which is essentially the most unstable CAPE (MUCAPE; see Table 1). Since there are numerous ways to derive CAPE, e.g. using different departures levels and (not) applying the virtual temperature correction (Doswell and Rasmussen, 1994), we do



not compare our CAPE values with other studies. Rather we look at CAPE differences across the cold front using the same
  consistent definition of CAPE. Figure 4 shows that pre-surface-frontal cells have the highest mean CAPE; up to around 650
  J kg⁻¹ 700-750 km ahead of the front. In comparison, post-700-frontal cells occur in environments of lower CAPE; between
  150–250 J kg⁻¹ on average. The mean CAPE at convective cell grid points is between 5–8 times higher than non-cell regions
  depending on the front relative region and is significant at the 95% confidence level. The difference between cell grid points
and non-cell regions is particularly large pre-surface frontal. Like the moisture variables the cell region means are between the
  non-cell region and cell grid point means.

### 3.3 Convective Inhibition

Environments with high convective inhibition (CIN) require stronger lifting so that parcels can reach their LFC. CIN is con-
sidered using parcels with different departure levels: most unstable parcel (MUCIN), mixed-layer parcel (MLCIN) and surface
based parcel (SBCIN). Due to the large quantity of data that would have needed to be requested only CIN at convective cell
  grid points is available for analysis. The reader is referred to section 2.4.1 for further details. Pre-surface-frontal cells form
  in environments with the strongest MUCIN, MLCIN and SBCIN with a mean of around -82 J kg⁻¹ of SBCIN (Figure 5).
  post-700-frontal cell environments had relatively weaker SBCIN in comparison (-15 J kg⁻¹). Therefore, more CIN needed to
  be overcome to initiate convective cells pre-surface-frontal compared to post-700-frontal. However, the stronger pre-surface-
255 frontal CIN may be advantageous for large CAPE build up (see Figure 4) since convection will not be initiated prematurely
  (Ludlam, 1980). MUCIN is lower than MLCIN and SBCIN for pre-700-frontal cells which could be explained by surface
  inversions (mostly at night and early morning) leading to higher CIN for parcels departing closer to the surface. Figure 5 is
  reproduced using daytime cells only (09–18 UTC) (Figure A1) showing that daytime pre-surface-frontal and post-700-frontal
  cells have a mean of -8 J kg⁻¹ and -28 J kg⁻¹ of SBCIN, respectively. A similar result was found for MLCIN and MUCIN.
While the overall CIN is weaker during the daytime, pre-surface-frontal CIN is still stronger than post-700-frontal during the
  daytime.

### 3.4 Quasi-geostrophic forcing for ascent

Q-vector convergence is a commonly used diagnostic by forecasters to highlight areas of geostrophic forcing for ascent or
descent. Q-vectors are derived from the quasi-geostrophic equations (Hoskins et al., 1978). The reader is referred to section
2.4.2 for further details on how Q-vectors are calculated. The 850 hPa level is the typical height of the boundary layer during
  the daytime and the location of the strongest capping inversion. Rising air parcels must overcome this inversion to reach the
  LFC. The 700 hPa and 500 hPa levels are also analysed to understand the importance of mid-level geostrophic forcing for
  ascent. Since large-scale lift is typically of the order of cm s⁻¹, it is unlikely sufficient to allow air parcels to reach their LFC
  (Trapp, 2013; his chapter 5.2). However, frontal simulations have indicated the lift associated with the cross frontal circulation
may be 1 order of magnitude higher (Koch, 1984; his Figure 7), thus may contribute to overcoming CIN. Synoptic-scale ascent
  can also steepen lapse rates and thus increase CAPE due to adiabatic cooling (Trapp, 2013; his Figure 5.2).





Figure 6 shows that cells marginally ahead of the surface front have the strongest convergence of the Q-vector at 850 hPa (hereafter QVEC850). The strongest QVEC850 convergence of non-cell region grid points is at the mean surface front location. The strongest QVEC850 divergence, which is linked to descending motion, is near-700hPa-frontal for all categories. However, near-700hPa-frontal the QVEC850 divergence is weaker at cell grid points compared to cell region grid points and non-cell regions. From 750 km behind the 700 hPa front up to near the surface front there is no significant difference between the QVEC850 for non-cell regions and cell regions indicating the importance of QVEC850 specifically where the convective cells are detected. QVEC700 convergence shows a similar result to QVEC850 with slightly weaker mean convergence and divergence. The regions of strongest ascent and descent shift to the left on the plots going from 850 hPa to 500 hPa due to the rearward slope of cold fronts with height.

Q-vector convergence at 500 hPa (hereafter QVEC500 convergence) is strongest near-700hPa-frontal across all three categories (Figure 6). Pre-700-frontal cells formed in environments with weaker QVEC500 than non-cell regions and cell regions, which is in contrast to QVEC700 and QVEC850. QVEC500 convergence at cell grid points near-700hPa-frontal (-50 to 50 km) is over 3 times stronger compared to non-cell regions and around 5 times stronger compared to the maximum cell grid point means at 700 hPa and 850 hPa. Non-cell regions post-700-frontal have mean QVEC500 divergence but cell regions and cell grid points have convergence of QVEC500. This result highlights the importance of upper-level forcing particularly on the development of convective cells particularly at the 700 hPa front and also post-700-frontal. The forcing for ascent could be linked to a post-frontal trough or may also act to destabilise upper-layers and hence increase CAPE.

## 3.5 Vertical Velocity

The vertical velocity at 850, 700 and 500 hPa ($w_{850\,hPa}$, $w_{700\,hPa}$ and $w_{500\,hPa}$) is shown from left to right in Figure 7. Like Q-vector convergence the variable can be used to highlight areas of ascending and descending motion. However, the vertical velocity is not solely linked to vertical motion due to geostrophic forcing for ascent or descent. Additional sources of ascending motion such as areas of convection may also show a signal in the vertical velocity variable in ERA5. At all levels the cell region and non-cell region grid point means are similar to the Q-vector in terms of where ascending and descending motion are present (Figure 6). However, there is a difference at cell grid points as there is mean ascending motion at all locations relative to the front at all three vertical levels. The strong anomaly at the 700 hPa front which was seen for QVEC500 is not seen for $w_{500\,hPa}$. The highest vertical velocity is at 500 hPa at cell grid points near the surface front (around 0.06–0.07 m s$^{-1}$). Ahead of the 700 hPa front vertical motion is maximised at 500 hPa; consistent with observations of quasi-geostrophic vertical motion being maximised around 500 hPa (Holton and Hakim, 2013). It is not clear whether the vertical velocity anomalies at cell grid points are related to convection being partially represented in ERA5 or relate to stronger large-scale lifting than climatology where convective cells were detected. The non-cell region grid points mean suggest that the lifting from the front at 850 hPa, 700 hPa and 500 hPa is maximised between the surface front and the 700 hPa front.





## 3.6 Solar heating

Solar heating (insolation) is linked to increased surface temperatures, which contributes towards atmospheric instability (Markowski
and Richardson, 2010; their Figure 7.9c). Solar heating also gives parcels positive buoyancy near the surface, which can help
parcels to be lifted to their LFC. The total incoming solar radiation (hereafter solar radiation) is shown in Figure 8. Only
timesteps between 09–18 UTC are used since solar radiation is weaker during the early hours of the morning, the late evening
and not possible during the night. The variable refers to the radiation accumulated during the hour prior to the ERA5 timestep,
thus before convective cell detection. Post-700-frontal cells develop with the largest solar incoming radiation (around 250 W
m$^{-2}$). The lowest solar radiation is ahead of the 700 hPa front but behind the surface front (50 to 150 km region), consistent
with where the total cloud cover is highest (Figure A2). Solar radiation at cell grid points between -250 to 50 km and pre-
surface-frontal is lower than at non-cell regions. The result may seem counterintuitive as solar radiation is thought to be a
driver of convective initiation so higher solar radiation would be expected before cell initiation. The result could indicate that
ERA5 struggles with the timing of convective initiation thus produces convective clouds before the actual convective initiation.
Nevertheless, on consultation of observational station data from the DWD (see section 2.4.3), the same negative anomaly was
observed for sunshine minutes (Figure A3). The negative anomaly is particularly strong pre-surface-frontal. One explanation
for the negative anomaly in solar radiation (ERA5) and sunshine duration (observational data) could be attributed to cloud
cover from pre-existing convective cells. The importance of cold pools (outflow boundaries) on convective initiation has been
already highlighted in previous literature. For example, Hirt et al. (2020) showed using high-resolution model simulations for
case studies that up to 50% of convective initiation is at the edge of cold pools. At the edge of cold pools it is likely to be cloudy
and convection may not be directly triggered by solar heating. This is especially true in the case of a mesoscale convective
system (MCS), where several new cells could initiate inside the pre-existing cloud system due to cell recycling.

## 3.7 Vertical Wind Shear

Vertical wind shear can both positively and negatively affect convective initiation (Peters et al., 2022). Numerical simulations
have shown that parcel buoyancy may be reduced in high-shear environments due to entrainment (e.g. Markowski and Richard-
son, 2010 and Peters et al., 2019). More recent work has shown that given updrafts meet an initial width and shear threshold
they can widen further, which in turn reduces their susceptibility to entrainment (Morrison et al., 2022 and Peters et al., 2022).

The bulk wind shear between the surface and 500 hPa is highest for convective cell grid points near-700hPa-frontal with
around 13 m s$^{-1}$ on average (Figure 9). The wind shear near-700hPa-frontal is also higher than non-cell region grid points by
around 1 m s$^{-1}$. There is less difference between cell regions and cell grid points near-700hPa-frontal. The wind shear decreases
at increasing distance away from the 700 hPa front. At a certain distance behind the 700 hPa front wind shear is lower at cell
regions and cell grid points compared to non-cell region grid points. While there is some asymmetry around the maximum,
post-700-frontal cells generally form in environments with comparable wind shear compared to pre-surface-frontal cells.
Since bulk wind shear between the surface and 500 hPa could potentially lead to a lower vertical distance being sampled for grid





points at higher elevation, the bulk wind shear between the surface and 6 km above ground level (AGL) as well as between the surface and 3 km AGL are also analysed at cell grid points using ERA5 model level data (Taszarek et al., 2023). Even though there are differences in the magnitude of the wind shear between the bulk wind shear surface–500 hPa (Figure 9) and 0–6 km AGL (Figure 10), the regions of maximum and minimum wind shear around the front are very comparable. Cells near the 700

hPa front have a particularly high 0–6 km AGL mean wind shear of around 23 m s⁻¹. Supercells have been shown in several studies to form in environments with around 20 m s⁻¹ of 0–6 km AGL of shear (e.g. Doswell and Evans, 2003). However, Pacey et al. (2023) showed that 2.5% of near-700hPa-frontal cells were associated with mesocyclones compared to around 5% of pre-surface-frontal cells (their Figure 11f). They showed also that post-700-frontal cells have an even lower fraction with mesocyclones where wind shear also remains high. Therefore, thermodynamics likely explain why a higher fraction of

pre-surface-frontal cells are associated with mesocyclones compared to near-700hPa-frontal and post-700-frontal cells owing to the more frequent overlap of high shear and high CAPE environments pre-surface-frontal.

### 3.8 Precipitation

To see how precipitation is represented in ERA5 across the front, large-scale and convective precipitation are shown in Figure 11. The large-scale precipitation is highest for all three categories between the 700 hPa and surface front. This is in agreement

with the classical conceptual model of an Ana cold front where the primary precipitation region associated to a cold front is behind the surface front (EUMeTrain, 2012). In this case, the primary precipitation region refers to mostly stratiform precipitation that results from condensation in the ascending warm conveyor belt. Figure 8 also shows that the lowest solar radiation is at this location relative to the front. Total cloud cover and high cloud cover for non-cell regions are also highest at this location (Figure A2) as well as vertical velocity at non-cell region grid points (Figure 7). At most regions relative to the front

the large-scale precipitation is higher at cell grid points compared to non-cell regions.

Convective precipitation is maximum 150–250 km ahead of the 700 hPa front for all three categories. This maximum is around 150 km behind the convective cell maximum found in Pacey et al. (2023) (see Figure 12). The differences could relate to weaker precipitation than the 46 dBZ threshold being more frequent closer to the front or ERA5 struggling to resolve the location of convection relative to the front. The minimum convective precipitation is 50–150 km behind the 700 hPa front

(again for all three categories) before increasing slightly further behind the front. The increase behind the 700 hPa front is consistent with the slight increase seen in the convective cell climatology (Figure 12). Convective precipitation at cell grid points is significantly higher compared to non-cell region grid points and shows a similar trend to the vertical velocity fields (Figure 7). This corroborates the hypothesis that the vertical velocity signal in Figure 7 comes from convection being partially represented in ERA5.

## 4 Discussion

A statistical comparison between cell grid points, cell regions and non-cell regions (section 2.3) depending on the region relative to the front has been made in section 3. How the results in this study relate to the cell frequency around the front is




discussed in this section. Figure 5a from Pacey et al. (2023) is shown here again as Figure 12. In this discussion, three front relative regions are focused on: pre-surface-frontal cells (300-750 km ahead of the 700 hPa front), near-700hPa-frontal (-50 to 50 km from the 700 hPa front) and post-700-frontal cells (-750 to -50 km from the 700 hPa front).

## 4.1 Pre-surface-frontal cells

Cells are between 4-5 times more frequent pre-surface-frontal than post-700-frontal and near-700hPa-frontal (Figure 12). A key result from this study is that while the environment quickly becomes unfavourable for cells behind the surface front, this is not the case in the other direction ahead of the front. For example, 750 km ahead of the 700 hPa front (around 450 km ahead of the surface front), Q-vectors remain convergent at 850 hPa and 700 hPa even when cells are not occurring (non-cell regions; Figure 6). On the other hand, 300 km behind the surface front (near-700hPa-frontal), Q-vectors at 850 and 700 hPa are divergent at non-cell regions and cell regions. Similar results are found for CAPE, surface dew points and solar radiation with the mean values remaining higher ahead of the surface front compared to the same distance behind the surface front. These results can explain why cell frequency remains high ahead of the front but sharply decreases towards the 700 hPa front. This is despite the fact that CIN is also higher for pre-surface-frontal cells (Figure 5). The presence of pre-surface-frontal convergence lines (Dahl and Fischer, 2016) is one possible source of lift that may aid parcels in overcoming this CIN.

## 4.2 Near-700hPa-frontal cells

Convection is least frequent surrounding the 700 hPa frontal line location (Figure 12). At 850 hPa, divergence of the Q-vector is typically found at this region relative to the front (Figure 6). This is also supported by the minimum vertical velocity at 850 hPa (Figure 7). Furthermore, the mean solar radiation is also lower and CAPE is lower than other regions relative to the front. The high wind shear (Figures 9 and 10) may also contribute towards the low cell frequency since deep moist convection may struggle to initiate when initial updraft width is low and shear is high (Morrison et al., 2022 and Peters et al., 2022).

## 4.3 Post-700-frontal cells

Post-700-frontal cells are associated with lower cell frequency than pre-700-frontal cells (Figure 12) and almost always occur during the daytime (Pacey et al., 2023; their Figure 6). Generally Q-vectors are divergent at 850, 700 and 500 hPa which would act to hinder the development of convective cells. Figure 3 shows that mid-level post-700-frontal relative humidity is generally low (50% or lower). Owing to the lower CIN (Figure 5) less lifting would generally be required to allow post-700-frontal cell initiation. However, parcels may be more susceptible to entrainment due to the combined effects of a dry mid-troposphere and wind shear thus not be able to reach the threshold of a convective cell (46 dBZ). These results can explain the overall low cell frequency post-700-frontal. The higher cell frequency compared to near-700hPa-frontal cells can be explained by solar radiation possibly acting as a source of lift and increasing instability (Figures 8 and A3).



## 5 Conclusions

In this study, the environments and lifting mechanisms associated with warm-season cold-frontal convective cells were analysed by combining automatic front detection and cell detection methods. Previous studies had primarily focused on the storm's environment looking at mid- and low-level moisture, instability and wind shear etc with less consideration of different lifting (triggering) mechanisms. Furthermore, variation in the environment and lifting mechanisms at different front relative regions had not been considered in any detail. Here, a comprehensive set of variables relevant for convective initiation were considered, including large-scale lifting, solar heating, CIN, CAPE, relative humidity and dew points. A strong focus was placed on differences in the convective cell environments and lifting mechanisms depending on the region relative to the front. How well each variable discriminates between regions where cells occurred and did not occur was also assessed. The primary findings of this study are highlighted below:

- Pre-surface-frontal cells form in the environments with the highest dew points and CAPE, around 16 C (Figure 2) and 600–700 J kg $^{-1}$ (Figure 4) on average. While cells at other front relative regions have lower CAPE and dew points, a significant positive anomaly exists compared to non-cell regions.

- Cells between the 700 hPa front and surface front form in environments with the highest mid-tropospheric relative humidity (Figure 3), which is linked to the frontal cloud band (Figures 8 and A2). Post-700-frontal cells had RH700 around 20–30% higher than non-cell regions, highlighting the importance of higher RH700 than usual to allow cell development and reduce the likelihood of entrainment.

- Pre-surface-frontal cells form in environments with the strongest Q-vector convergence at 850 hPa and 700 hPa (Figure 6) and also with the largest CIN (Figure 5).

- Strong large-scale lifting at 500 hPa is key for cell initiation near-700hPa-frontal (Figure 6). The lifting may not be (only) relevant for maintaining positive buoyancy of updrafts, rather (also) steepening lapse rates and increasing CAPE through adiabatic cooling. Conversely, upper-level forcing is less relevant for pre-surface-frontal cells.

- In the post-700-frontal region, non-cell regions have mean divergence of the Q-vector at 850, 700 and 500 hPa, whereas weaker divergence or even convergence of the Q-vectors is found at cell locations.

- Unlike the Q-vector fields, a strong positive anomaly for ascending motion in the vertical velocity is found for cells at all front relative regions.

- Solar radiation (ERA5) and sunshine duration (observational data) are generally lower at cell locations compared to non-cell regions, including post-700-frontal cells. While solar heating may be relevant for the first cell initiation, we speculate other factors (e.g. outflow boundaries) are more relevant for the majority of cold-frontal cell initiation.

These results advance understanding of the environments in which cold-frontal convective cells form and the importance of different lifting mechanisms on cell development depending on the region relative to the front. The results from this study



leave several interesting open questions for future work. For example, is the upper-level lifting generally more important for increasing instability through adiabatic cooling or also relevant for maintaining positive buoyancy? The positive vertical velocity anomaly at cell grid points which is not present in the Q-vector fields raises questions on whether the signal originates from ERA5 partially representing cell initiation.

*Code and data availability.* ERA5 data can be downloaded from the Copernicus servers (2020). KONRAD is available for research purposes on request (contact kundenservice@dwd.de). Front detection and plotting code are available on request from the corresponding author.

*Author contributions.* GP carried out the data analysis and wrote all sections of the manuscript. SP and LS provided comments and support during the data analysis and manuscript process. SP and LS wrote the original research proposal.

*Competing interests.* At least one of the (co-)authors is a member of the editorial board of Weather and Climate Dynamics.

*Acknowledgements.* This research has been funded by Deutsche Forschungsgemeinschaft (DFG) through grant CRC 1114 'Scaling Cascades in Complex Systems, Project Number 235221301', Project C06 "Multi-scale structure of atmospheric vortices". We thank the German Weather Service (DWD) for providing the KONRAD cell detection and tracking dataset. The Open Data Server from the DWD also allowed sunshine duration data to be downloaded. We thank Johannes Dahl for useful discussions and insights as well as Mateusz Taszarek for providing convective parameter data which was used in sections 3.3 and 3.7.



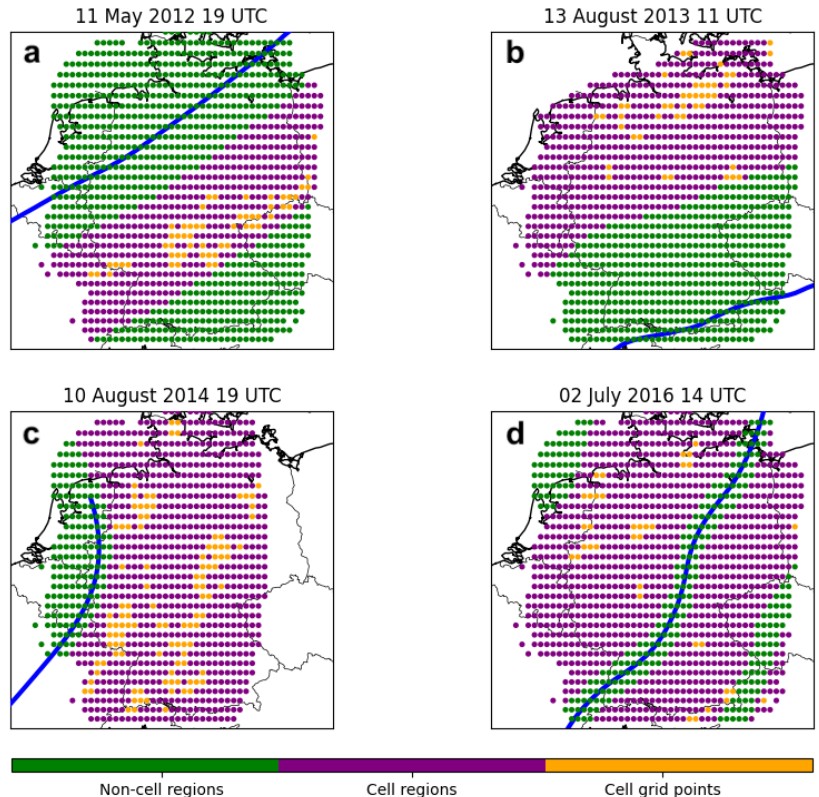

**Figure 1.** Visualisation of how cell grid points (yellow), cell regions (purple) and non-cell regions (green) are defined for four timesteps on different days. The 700 hPa frontal line is shown by the blue line and was detected using the methods defined in section 2.1.




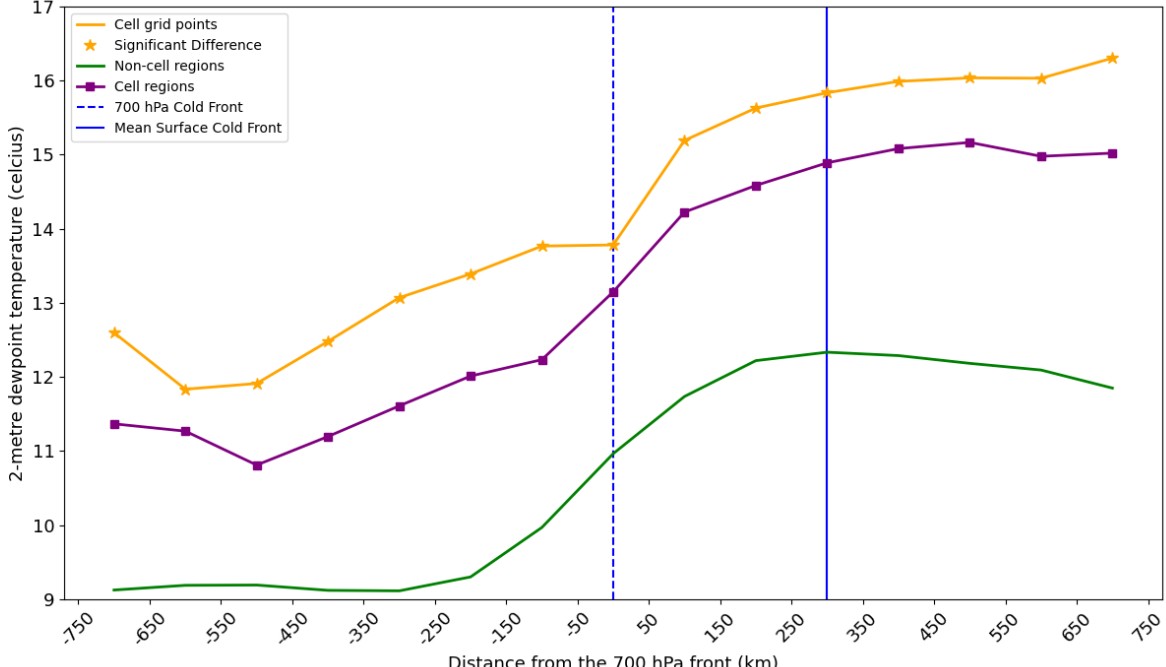

**Figure 2.** 2-metre dewpoint temperature (celcius) depending on distance from the 700 hPa front (km) for cell grid points (orange), cell region grid points (purple) and non-cell region grid points (green). Stars indicate that the convective cell grid point mean is significantly different from the non-cell region grid point mean at the 95% confidence level based on a Welch's t-test, which does not assume equal population variance. The dashed vertical line and solid vertical line represent the 700 hPa front and mean surface front location (see section 2.1.1), respectively.



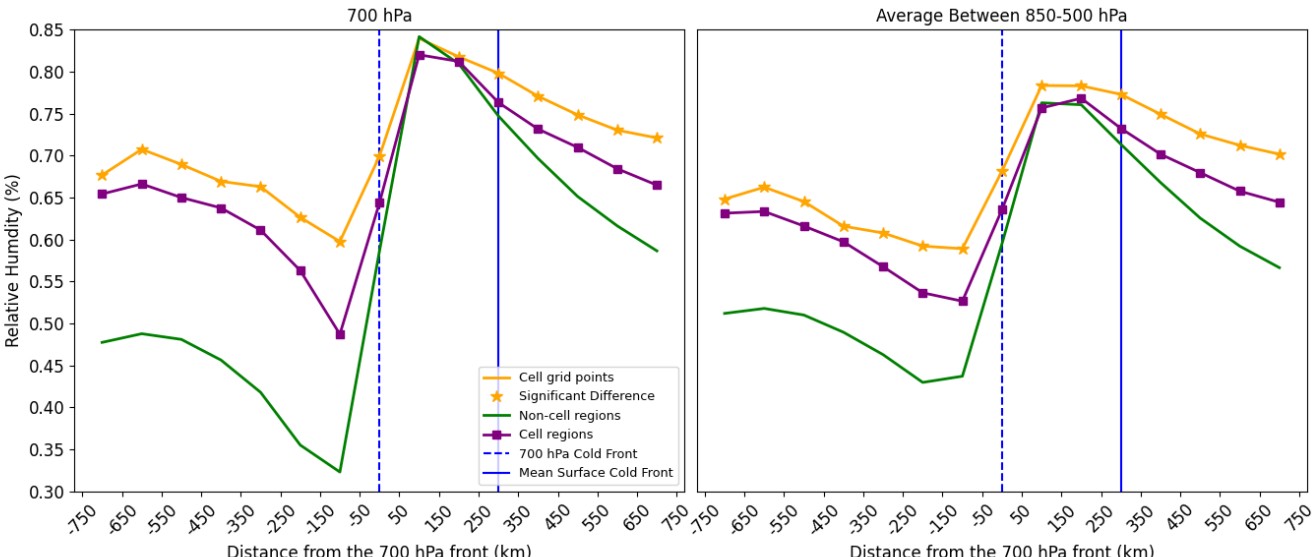

**Figure 3.** As Figure 2 but for relative humidity at 700 hPa (%) and mean relative humidity between 850–500 hPa.

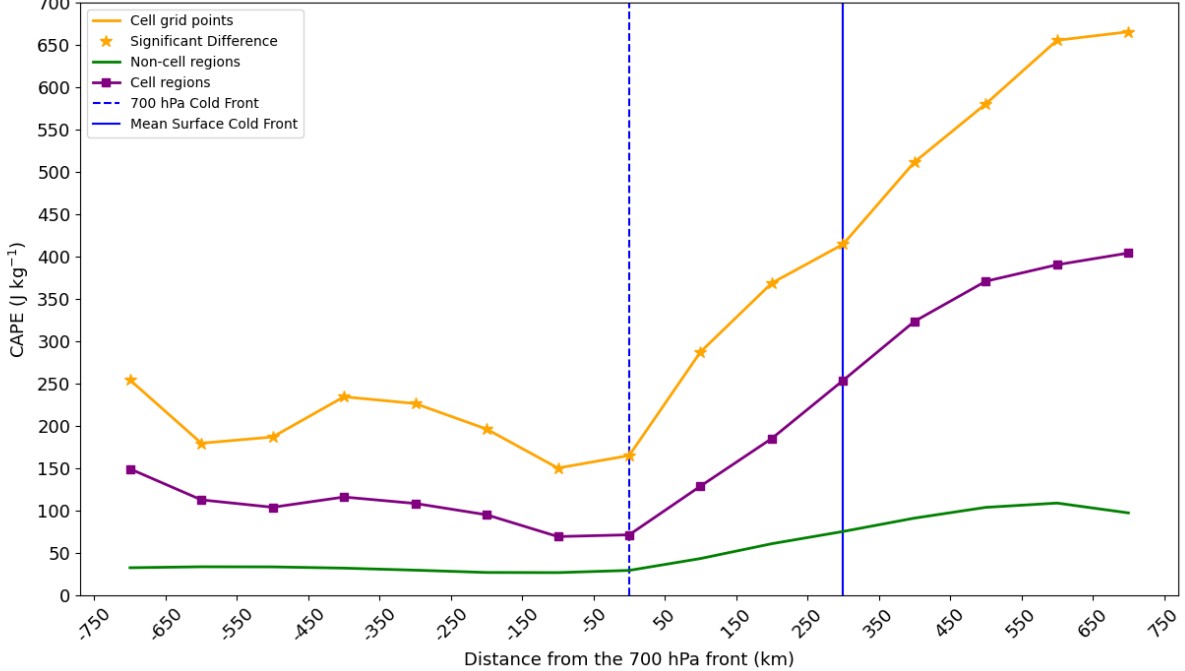

**Figure 4.** As Figure 2 but for CAPE. The ERA5 CAPE variable uses the parcel with the highest CAPE considering different departure levels below 350 hPa.





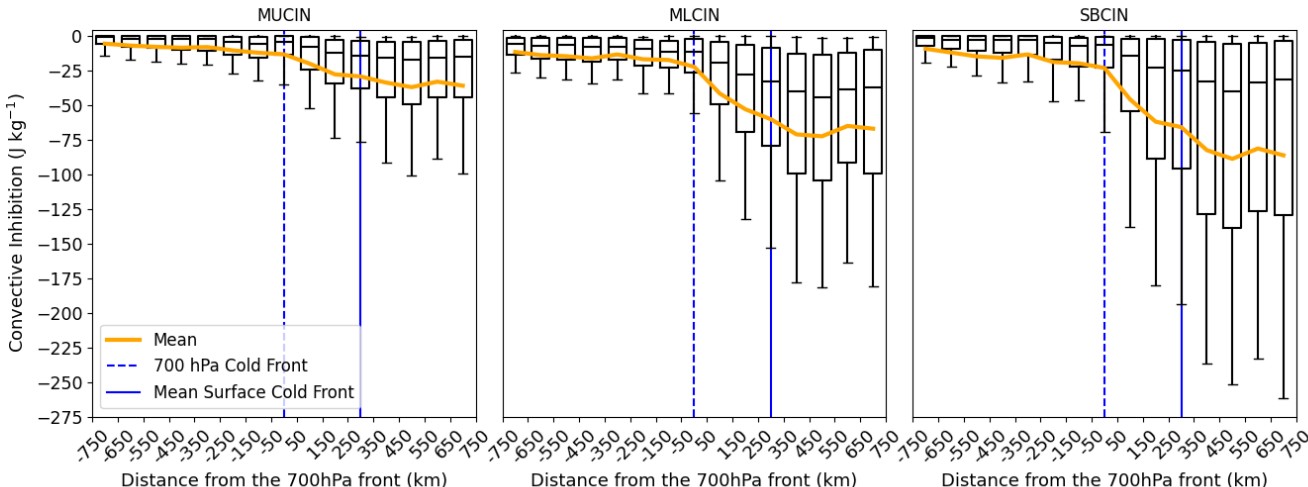

**Figure 5.** MUCIN, MLCIN and SBCIN (J kg$^{-1}$) for convective cell grid points only. The whiskers represent the 10th and 90th percentiles, the median is represented by the horizontal black line and the mean by the orange line.

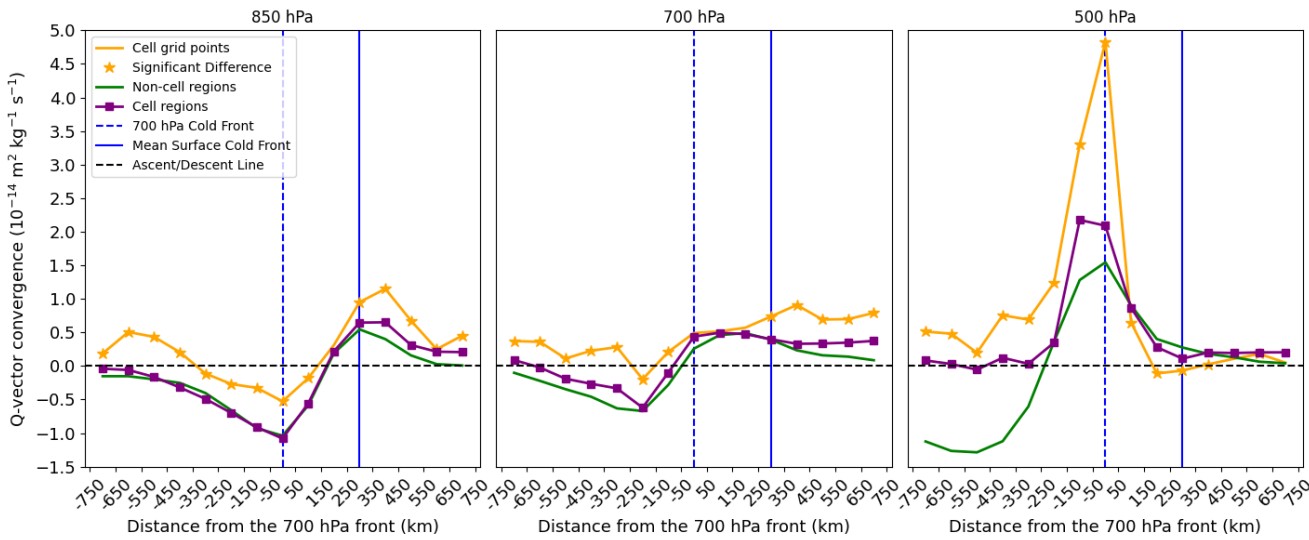

**Figure 6.** As Figure 2 but for Q-vector convergence at 850 hPa, 700 hPa and 500 hPa (left to right). Positive and negative values indicate convergence (ascending motion) and divergence (descending motion) of the Q-vector, respectively. Q-vectors are derived using the methodology described in section 2.4.



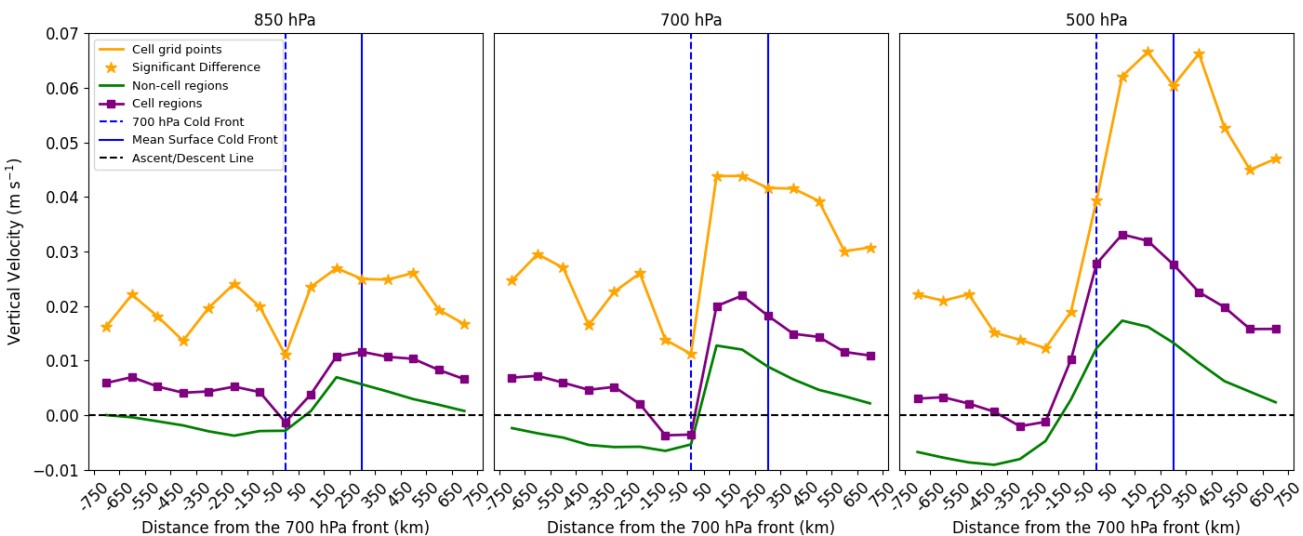

**Figure 7.** As Figure 2 but for vertical velocity at 850, 700 and 500 hPa. Postive and negative values indicate ascending and descending motion respectively.

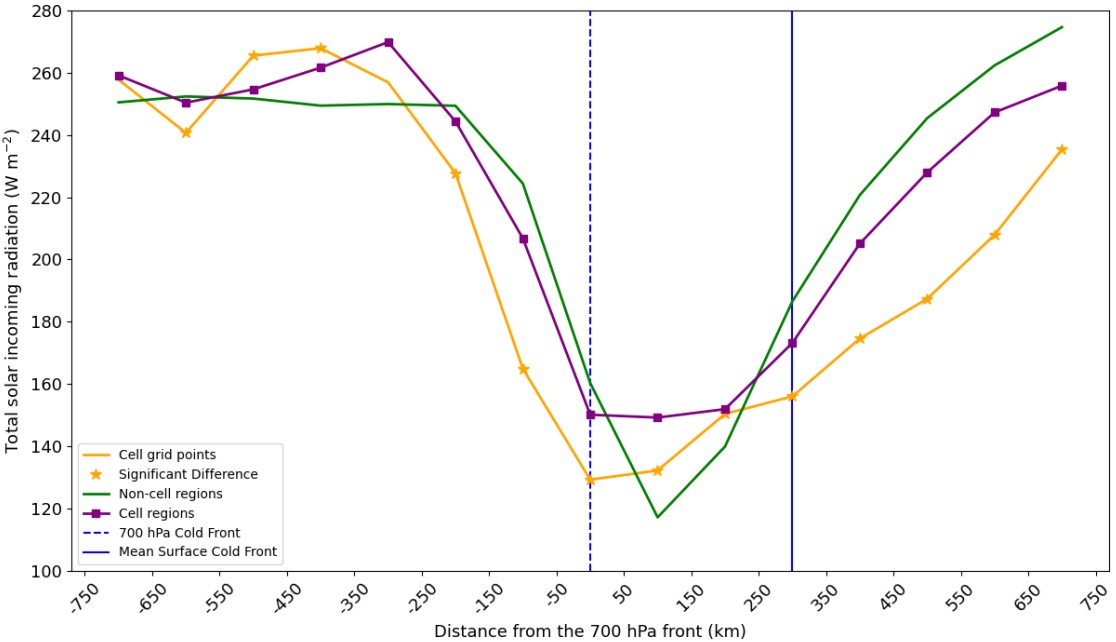

**Figure 8.** As Figure 2 but for total incoming solar radiation (W m$^{-2}$) only using timesteps between 09–18 UTC.





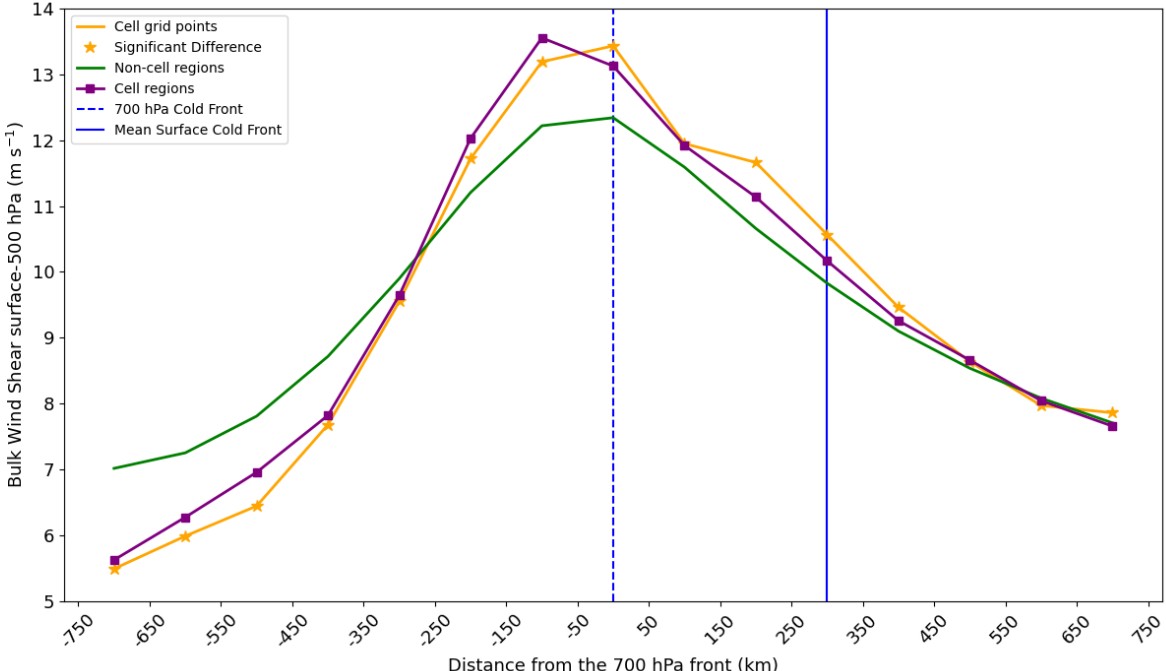

**Figure 9.** As Figure 2 but for wind shear between the surface and 500 hPa.



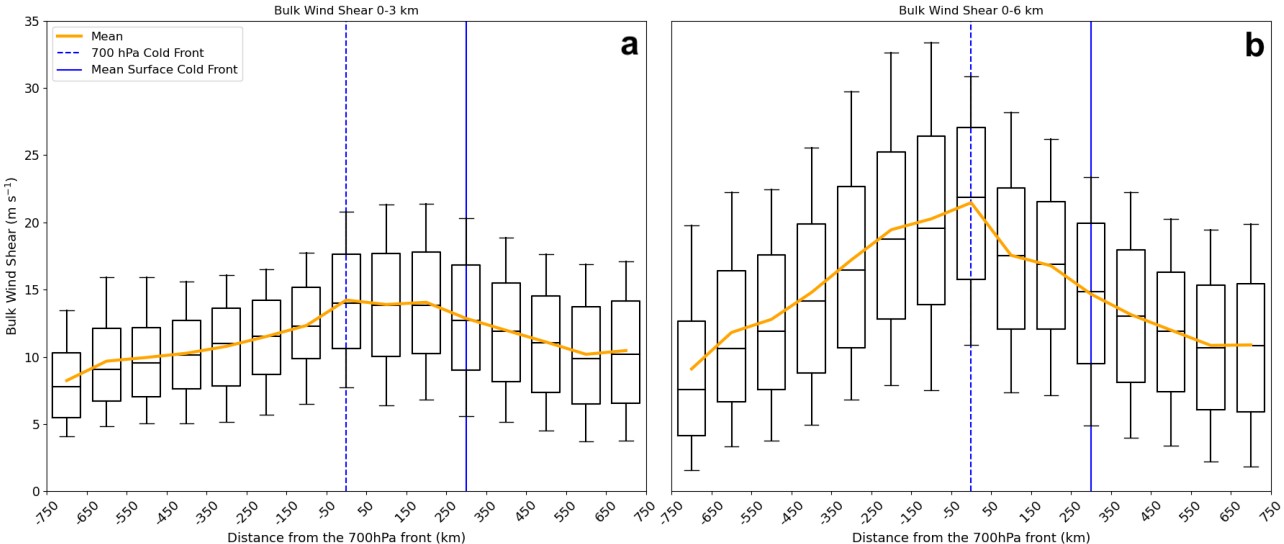

**Figure 10.** As Figure 5, but for bulk wind shear between the surface and 3 km AGL (a) and between the surface and 6 km AGL (b). Only convective cell grid points are shown.

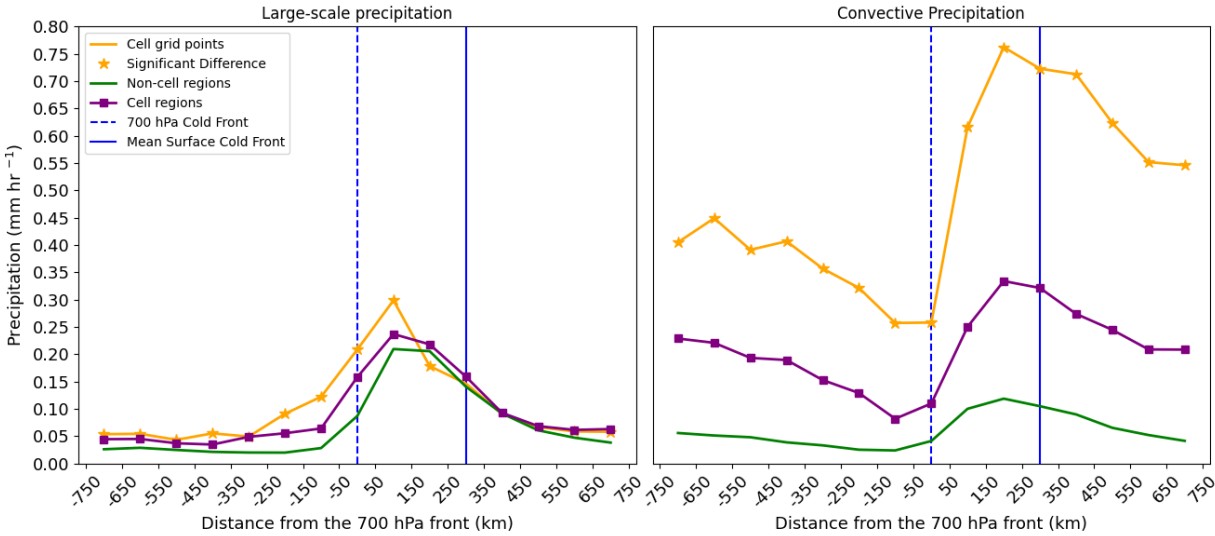

**Figure 11.** As Figure 2 but for large-scale precipitation and convective precipitation.



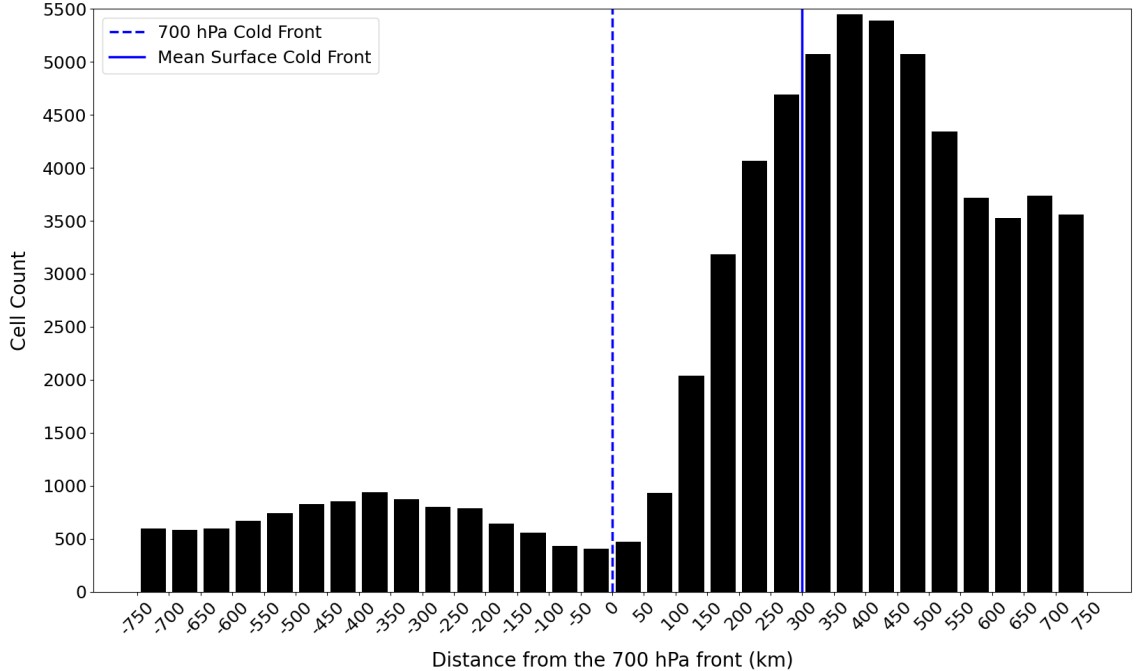

**Figure 12.** Convective cell frequency depending on the distance from the 700 hPa front (adapted Figure 5a from Pacey et al., 2023).



**Table 1.** A list of variables analysed in this study and the associated level and dataset.

| Variable | Level | Dataset | Units |
|---|---|---|---|
| Dewpoint temperature | 2-metres above ground level | ERA5 | K |
| Relative Humidity | 700 hPa, 850-500 hPa average | ERA5 | % |
| CAPE[1] | single level | ERA5 | J kg$^{-1}$ |
| Convective Inhibition (CIN) | different departure levels | ERA5[2] | J kg$^{-1}$ |
| Q-vector convergence | 850 hPa, 700 hPa, 500 hPa | ERA5 | m$^2$ kg$^{-1}$ s$^{-1}$ |
| Vertical Velocity | 850 hPa, 700 hPa, 500 hPa | ERA5 | m s$^{-1}$ |
| Total incoming solar radiation | surface | ERA5 | W m$^{-2}$ |
| Sunshine duration | surface | DWD Station Data | minutes |
| Vertical Wind Shear | surface-500 hPa, 0–3 km, 0–6 km AGL[3] | ERA5 | m s$^{-1}$ |
| Large-scale precipitation | surface | ERA5 | mm hr$^{-1}$ |
| Convective precipitation | surface | ERA5 | mm hr$^{-1}$ |

[1]The ERA5 CAPE parameter downloaded from the Copernicus Climate Data Store (Hersbach et al., 2018b) is used. CAPE is derived considering parcels departing from different model levels below the 350 hPa level and the departure level with the highest CAPE is retained. In essence, the ERA5 CAPE parameter is the Most Unstable CAPE (MUCAPE).

[2]The data source and methods to calculate CIN are shown in section 2.4.1.

[3]Above ground level (AGL).





**Table 2.** Number of grid points in each category (non-cell regions, cell regions and cell grid points).

| Distance from front | Non-cell regions | Cell regions | Cell grid points | Total |
|---|---|---|---|---|
| -750 to -650 km | 375,897 | 33,165 | 1,230 | 410,292 |
| -650 to -550 km | 403,822 | 38,920 | 1,571 | 444,313 |
| -550 to -450 km | 428,715 | 49,085 | 1,942 | 479,742 |
| -450 to -350 km | 458,274 | 52,914 | 2,120 | 513,308 |
| -350 to -250 km | 483,334 | 49,239 | 1,975 | 534,548 |
| -250 to -150 km | 512,047 | 42,658 | 1,753 | 556,458 |
| -150 to -50 km | 539,207 | 39,002 | 1,128 | 579,337 |
| -50 to 50 km | 522,995 | 38,111 | 1,035 | 562,141 |
| 50 to 150 km | 530,638 | 88,049 | 3,301 | 621,988 |
| 150 to 250 km | 516,253 | 134,347 | 7,863 | 658,463 |
| 250 to 350 km | 503,009 | 166,937 | 10,348 | 680,294 |
| 350 to 450 km | 505,580 | 167,999 | 11,275 | 684,854 |
| 450 to 550 km | 525,694 | 147,972 | 9,524 | 683,190 |
| 550 to 650 km | 555,236 | 123,031 | 7,571 | 685,838 |
| 650 to 750 km | 556,448 | 108,553 | 7,072 | 672,073 |



## Appendix

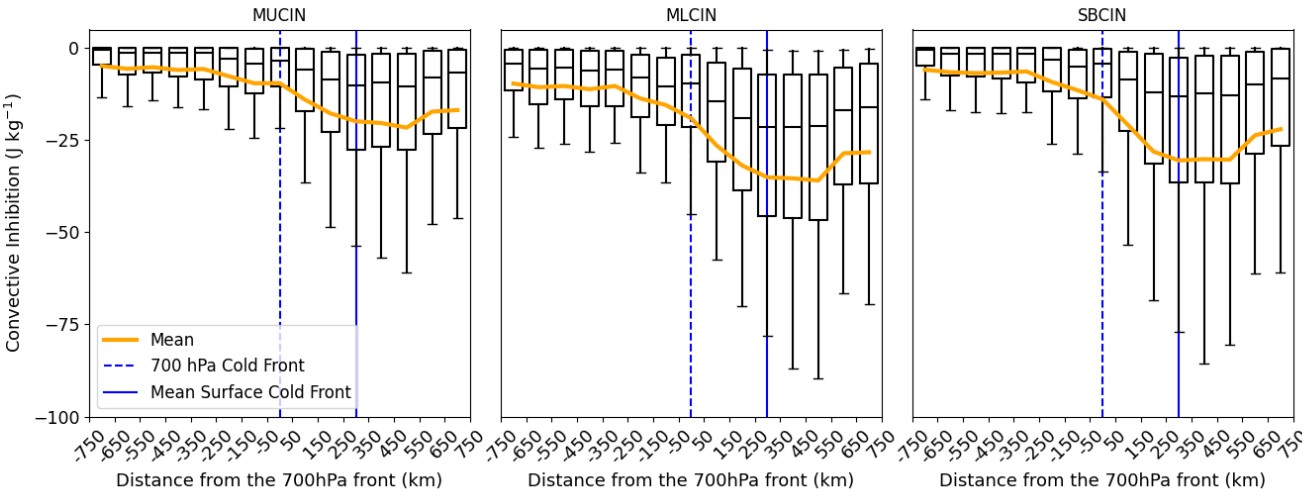

**Figure A1.** As Figure 5 but for cells between 09–18 UTC only. Note the smaller y-axis range compared to Figure 5.

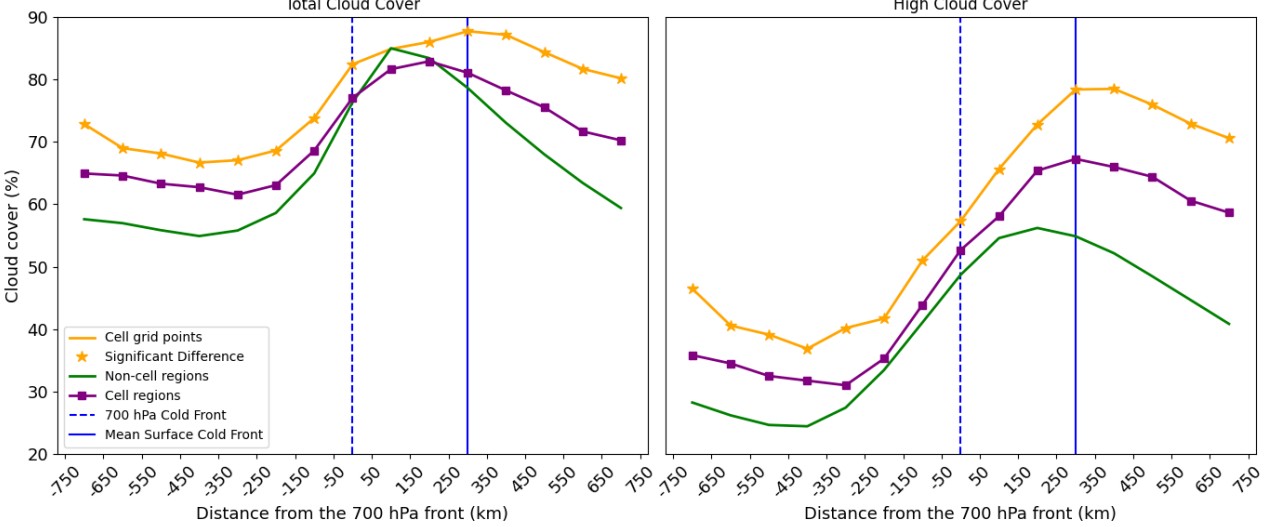

**Figure A2.** As Figure 2 but total cloud cover (left) and high-cloud cover (right).



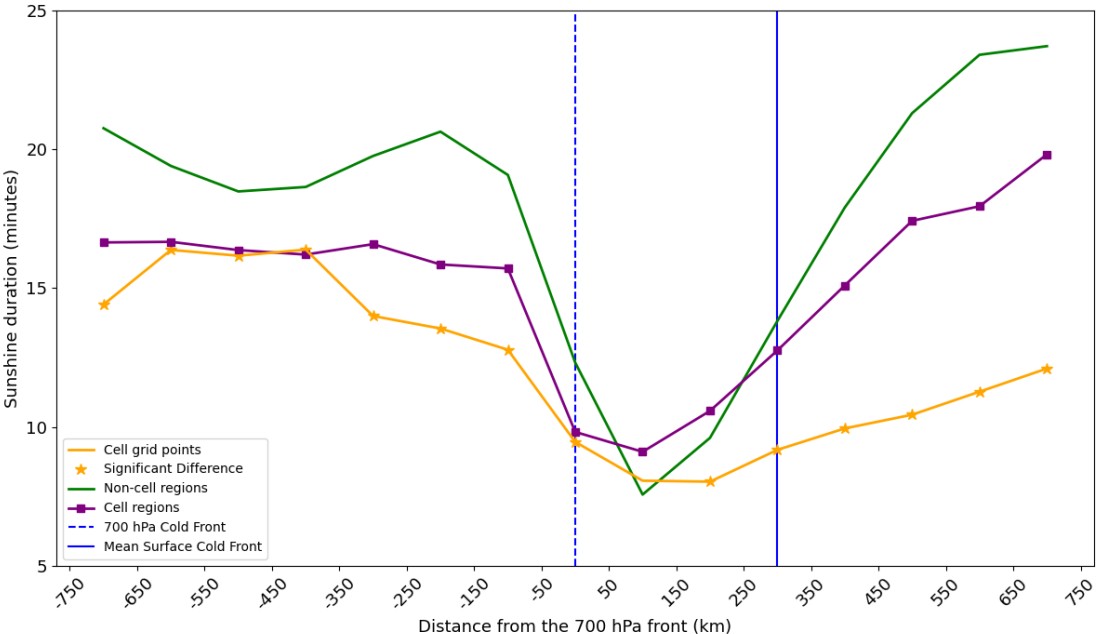

**Figure A3.** As Figure 2 but for sunshine duration (minutes) only using timesteps between 09–18 UTC. Observational data is used in this figure as described in section 2.4.3.



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
