# Peer review of "Environments and lifting mechanisms of cold-frontal convective cells during the warm-season in Germany"

_EGUsphere, 2024_

## Referee Comment (RC1)

Review of egusphere-2024-2978

*'Environments and lifting mechanisms of cold-frontal convective cells during the warm-season in Germany"*

by George Pacey et al.

Manuscript in review for Weather and Climate Dynamics

**Summary**

To characterize the environments in which convective cells in cold frontal regions form, the manuscript combines convective cell tracking and detection with automatic front detection. The thermodynamic conditions and different lifting mechanisms for (i) convective cells, (ii) cell environments, (iii) and non-cell environments are compared using ERA5 data. Moreover, the results are binned in different categories based on their distance to the cold front location which facilitates to discriminate between post-frontal, frontal, and pre-frontal convection. Their results indicate that environmental conditions favorable for pre-surface-frontal and post-frontal convection differ. For example, pre-surface frontal convection appears to be favored by large CAPE values, while for post-frontal cells mid-level relative humidity appears to be relevant for convection to be present. The manuscript addresses open questions concerning the initiation of convection near cold fronts, and the applied methods are suitable to address the research questions. Generally, the manuscript is well-structured and the figures are presented clearly. The research questions are clearly stated and of relevance for the readership of *Weather and Climate Dynamics*. I recommend publication of the manuscript in *Weather and Climate Dynamics* but I have several comments that should be addressed prior to publication, which are outlined in detail below.

**1 General comments**

1 General structure of the manuscript
Overall, the manuscript is well-structured. Yet, the Results Chapters include individual phrases and paragraphs that rather belong in the Introduction or Methods or Discussion section. I would ask the authors to make sure that the information is consistently placed in the appropriate sections and revise/streamline the Results Chapter accordingly. This does not pertain to the overall structure of the manuscript (which I think is good) but rather to several individual sentences/paragraphs. For example, Section 3.4 starts with one paragraph of introduction to Q-vector convergence which could be substantially shortened by moving the content to the Introduction and/or Methods Sections. Similarly, I would ask the authors to make sure that their discussion of results is consistently placed in the Discussion Section (and not in the Results Section).

2 Introduction of data set
The presented study follows up on a study published by Pacey et al.

(2023). I think the reader would profit from summarizing the key results from the predecessor study (Pacey et al., 2023) in more detail.

3 Quantification of uncertainty
Most figures show only mean values binned in distances from the front. I would appreciate if a measure of uncertainy (e.g., standard deviation as shading, percentiles, etc.) could be additionally shown to illustrate the associated uncertainty and variability. I appreciate that significant differences are emphasized in the figures, however, including a measure of uncertainty would be beneficial, for example, as shown by the boxplots in Fig. 5. Figure 5 also shows that the distributions are not Gaussian, i.e., it may be useful to show the median in addition to the mean for some variables that are not normally distributed.

4 Definitions of CIN (vs CAPE)
The authors apply only one (of many definitions) for CAPE but several definitions of CIN. I would ask the authors to better justify why different definitions are used for CIN (but not CAPE). I'm not sure the manuscript profits substantially from the other definitions of CIN (i.e., the authors could consider moving panels (b) and (c) of Fig. 5 to the Appendix to streamline the manuscript), as the key conclusions do not depend on the detailed definition of CIN. Instead, I would be interested to see a comparison of the CIN distribution of convective grid points with the CIN distribution of the convective environment (and non-convective environment). Yet, the authors have mentioned that the computation of CIN for that many grid points is computationally demanding, and I understand if this is not feasible here.

5 Conclusions
In the conclusions, the authors summarize the most relevant factors for convective initiation by discussing each variable separately (e.g., CAPE, RH, Q-vector convergence, etc.) following the structure in the Results Section. I would find it more helpful if the authors could cluster the key factors and differences for pre-surface-frontal, near-700hPa-frontal and post-700-frontal as substantial differences between these categories are presented (i.e., similar to the structure in the Discussion Section). Finally, the authors have clearly defined three research questions in the introduction, which could be addressed again in the conclusions.

6 Writing style
This is only a minor point, but I would ask the authors to capitalize the first letter if a specific section or chapter is referenced (e.g. Section 1, instead of section 1). Moreover, please double-check the usage of ";" throughout the text (see also specific comments below).

**2 Specific comments and technical corrections**

1. l. 5 "At other front relative regions": At this point in the abstract, it is unclear what "other front relative regions" refers to. Please rephrase or add additional explanations in the abstract.

2. **Introduction**

3. l. 15: "; primarily due to convective permitting models (CPMs) at increased resolution": This does not read like a complete sentence. I would suggest to avoid the usage of ";" in the text (please also check later occurrences in the manuscript). I also think that it is common to use "convection-permitting" (instead of "convective permitting").

4. l. 28: Please remove "etc" and specify.

5. l. 84: Please avoid double-brackets ") (". See also l. 129, 258, 357.

6. l. 32 ff: This reads a bit colloquially, please rephrase.

7. l. 43 f: "The literature would benefit from studies quantifying the relevance of frontal lifting at different regions relative to the front, especially during the warm-season.": This sentence appears a bit out of context, I'm not sure if it necessary here.

8. l. 51 ff: The role of wind shear should be discussed together with other factors relevant for convection, i.e., in the paragraph starting in l. 27. Moreover, following General Comment 1, I would ask the authors to include the background information provided in the Results sections in the introduction. Overall, the introduction would benefit from more clearly summarizing and structuring key variables relevant for convection, targeting specifically convection embedded in the frontal environment.

9. **Methods**

10. l. 85: I would ask the authors to include one explanatory sentence on the TFP equation.

11. l. 106: "The process is repeated 30 times to remove any local-scale features.": Please explain here, why specifically "30 times" was chosen. I would ask the authors to elaborate on why this specific smoothing method was applied (in contrast to other methods)?

12. l. 109: "than some previous studies": Please include those studies here.

13. l. 118 f: "This is also supported by the mean maximum climatological surface convergence in ERA5 data (Pacey et al., 2023; their Figure 3)". Please add some additional information on how this figure supports the statement, such that the reader does not have to read the mentioned publication themselves.

14. l. 132 ff. Please label the criteria following your approach above (i.e., "(A)", "(B)", etc.), and add text describing the criteria.

15. l. 142: Does the time of first detection correspond to the first exceedance of 46 dBZ? Would it be possible to track these cells also before they reach maturity, i.e. in their developing phase with lower reflectivity?

16. l. 139 ff: "Since some cells have a lower area than the grid size the bounds are increased by 0.125 degrees (half a grid point)": I find this difficult to understand, please rephrase.

17. Figure 1: This figure nicely illustrates the definition of the three categories. For further illustration, the authors could additionally include the position of the surface cold front as well as the defined pre-frontal, pre-surface frontal and post-frontal regions.

18. l. 152, Table 2: I appreciate that detailed numbers of grid points in each category are provided in Table 2, yet it would be easier to comprehend if the numbers were shown as an additional figure.

19. l. 161 ff: The choice of variables should clearly be motivated in the introduction, such that refering to literature in this short paragraph is not required anymore.

20. l. 166: As mentioned above, it is not fully clear to me why different definitions of CIN, but not of CAPE are used.

21. l. 168: "So that a CIN value is present for all grid points": Please rephrase (e.g., "To ensure that ...").

22. Equation 3: Please define the symbols/abbreviations in the text.

23. l. 193 f. I would ask the authors to elaborate on the smoothing method and how the number of smoothing cycles was exactly determined (see also previous comment above)?

24. **Results**

25. l. 203: It could be helpful for the reader to show and discuss the occurrence frequency of convection (Fig. 12) before Sub-sections 3.1, 3.2, etc. to familiarize the reader with the dataset.

26. l. 207 ff: The authors emphasize the differences in dewpoint between convective and non-convective environments. Did the authors also consider the 2-m temperature distributions? Are the differences in dewpoint related to differences in humidity (i.e., dew point depression) or to differences in the background temperature? I would be interested to see Fig. 2 for 2-m temperature.

27. l. 236-241. This is repetitive. Please streamline and/or move to Methods and Discussion Sections.

28. l. 241 f: Please rephrase and avoid using ";".

29. l. 259: I assume the two numbers have been swapped? I guess post-700-frontal should have lower absolute CIN values.

30. l. 262: "by forecasters" is not necessarily required.

31. l. 263-271: Please streamline this paragraph.

32. l. 279: "shift to the left on the plots": Please rephrase.

33. l. 297 f: The authors show mean vertical velocity for cell grid points of on average a few cm/s. I would ask the authors to relate those numbers (in the Discussion) to typically observed convective updraft velocities and discuss its implications for using ERA5 vertical velocity for studying convection. Is vertical velocity normally distributed and are large outliers present?

34. Caption Fig. 7: Please correct the typo: "Postive"

35. l. 335-336: This sentence is not fully clear to me, please rephrase.

36. l. 351: I would appreciate if the authors could include a more original reference (in addition to EUMeTrain).

37. l. 352: Please rephrase "condensation" by "cloud formation" or similar, as condensation inherently implies that precipitation only forms from warm-phase cloud processes.

38. l. 354 ff: Fig. A2 suggests that a substantial number of identified cells occurs in an environment with large cloud cover, and thus, may be embedded in a larger precipitating cloud system (see Fig. 11), such as the warm conveyor belt (which has also been mentioned). The Discussion Section could profit from including studies on precipitation characteristics and distribution in the warm conveyor belt airstream compared to pre-frontal convection.

39. l. 361-364: I'm not sure if I agree with the conclusions in this paragraph. Assuming ERA5 would (at least partially) represent convection, the precipitation signal would show up in "Large-scale precipitation", and not in the parameterized convective precipitation. In this case, I would expect a difference between the three categories in Fig. 11a (which is very small, in particular pre-frontal). Instead, the parameterized "convective precipitation" differs between categories (Fig. 11b), suggesting that convection cannot explicitly be represented in ERA5 (and needs to be parameterized). Apologies, in case I mis-understood this paragraph. I would ask the authors to rephrase this paragraph.

40. l. 366 ff: Please streamline this paragraph, this information has been repeated several times.

41. l. 371 ff: I appreciate that the authors discuss and summarize the relevant factors for (i) pre-surface-frontal cells, (ii) near-700hPa-frontal cells, and (iii) post-700-frontal cells. While the overall structure is good, this section could profit from (even more thoroughly) comparing the presented results to previous studies, and e.g., pick up literature that has been mentioned in the introduction.

42. l. 408: typo: include "°" in "16 C".

43. **Conclusions**

44. l. 401: Please remove "etc" and specify.

45. l. 408-426: In general, I think it is ok to use bullet points to emphasize key conclusions. Yet, I would reduce the number of bullet points and more strongly aggregate the relevant information.

46. l. 422 f: It is expected that convective cells are associated with positive vertical velocity. In Section 3.5, the authors have briefly mentioned the implication for consistently seeing this signal in ERA5 data. I think the manuscript would profit from a more thorough discussion on ERA5 and its ability to (at least partially) represent convection and convective precipitation. The last sentence of the manuscript brings up this open question, yet it could be discussed more thoroughly.

47. Have the authors considered including a schematic that illustrates and summarizes the relevant factors that discriminate between convective and non-convective conditions in different regions of the front?

---

## Referee Comment (RC2)

**Review comments**
**egusphere-2024-2978 (WCD)**

**Title:** Environments and lifting mechanisms of cold-frontal convective cells during the warm-season in Germany

**Authors:** Pacey et al.

**Recommendation:** minor revisions

**General**

By using automated front detection and convective cell tracking data, this study analyzes the environmental conditions and lifting mechanisms influencing convection near cold fronts in warm seasons in Germany. Results show that pre-surface-frontal cells tend to form in areas with the highest surface dew points and CAPE. Other front-relative regions also support cell formation, though with lower CAPE and dew points compared to non-cell regions. Mid-level humidity helps distinguish post-frontal cell locations from non-cell regions. Pre-surface-frontal cells experience strong large-scale lifting at 850 hPa and 700 hPa, with high convective inhibition. Significant large-scale lifting also appears post-frontal, especially at 500 hPa. Additionally, less sunshine was observed before cell initiation compared to non-cell regions, suggesting that solar heating may not drive most cold-frontal cell initiation.

This study provides a valuable analysis of the atmospheric conditions that influence convection near cold fronts. I find the subject interesting and well within the scope of WCD. It is well written, has a clear structure, and good illustrations. However, I have a couple of minor concerns which can maybe be solved with more explanations. I expand on some of these concerns below and outline additional minor/technical comments.

**Specific comments**

- The authors study the warm-season in Germany, but only convection related to frontal zones. I assume that a large part of convection occurs in situations with weak synoptic forcing as well. Can the authors comment on the overall relevance of frontal convection in Germany? Moreover, several field experiments were carried out to better understand convection initiation in Germany. At least some of those with their main findings should be cited.

- Some general information about convection initiation is given in the introduction. However, the multiple effects of mountains or land-surface heterogeneities are not mentioned. I recommend enlarging that section with these points, particularly the role of low-level convergence zones.

- One aspect I may have missed in the manuscript is the fact that even if CAPE values are lower in post-frontal regions, the atmosphere is often unstable due to the advection of colder air at higher levels. Warming by solar radiation is then often sufficient to initiate convection in large areas.

- I am a bit concerned about the fact that pre-surface-frontal cells form in environments with the highest CAPE AND strongest CIN. High CAPE values usually occur at low altitudes of the LFC which also often is associated with smaller values of CIN. Please comment on this.

- The authors detect and track convective cells based on radar data from the DWD. Radar-derived precipitation adjusted to surface observations is also available from the DWD on the same domain. I wonder why this data set is not used for precipitation? Precipitation and in particular convective precipitation is certainly not best represented in the ERA5 data set.

- Why do you need a smoothing of $\theta_e$ and why 30 times?

- P6: You state that cold fronts reach the southern parts of Germany less frequently. Is this related to the more complex terrain there?

- P6, L162: You state that wind shear affects convective initiation. Wind shear, however, is not a trigger mechanism, it is decisive for the evolution and organisation of the initiated convection. I suggest to write that it affects convection and not convective initiation.

- As for convective precipitation, I doubt that the vertical velocity in the ERA5 data set is really representative for deep convection. Please comment.

- What are the implications for forecasting convective storms near frontal zones? Are there any ways to improve numerical models with these findings?

**Technical comments**

- P3, L85: Paramter → Parameter

- Line breaks occur between numbers and their units troughout the entire manuscript (e.g. P4, L117-118). Please correct that everywhere.

- P6, L151: ...bin at  the current...

- P6, L165: A full list...  **is** shown...

- P6, L177: The quasi-geostrophic forcing for ascending and descending motion can be **measured** using the Q-vector convergence... I think "measured" is not the best word here as this is not a measurement. Maybe "expressed" or "described" are better options.

- P10, L286: This result highlights the importance of upper-level forcing **particularly** on the development of convective cells **particularly** at the 700 hPa front and also post-700-frontal. Please rephrase.

- P14, L408: 16 C → 16°C

- P17, Fig. 2 caption: celcius → Celsius

- P20, Fig. 7 caption: Postive → Positive

- P29, L509: 1.  edn.

---

## Author Comment (AC1)

**Response to RC1**

*Authors' responses are in red italics*

**Summary**

To characterize the environments in which convective cells in cold frontal regions form, the manuscript combines convective cell tracking and detection with automatic front detection. The thermodynamic conditions and different lifting mechanisms for (i) convective cells, (ii) cell environments, (iii) and non-cell environments are compared using ERA5 data. Moreover, the results are binned in different categories based on their distance to the cold front location which facilitates to discriminate between post-frontal, frontal, and pre-frontal convection. Their results indicate that environmental conditions favorable for pre-surfacefrontal and post-frontal convection differ. For example, pre-surface frontal convection appears to be favored by large CAPE values, while for post-frontal cells mid-level relative humidity appears to be relevant for convection to be present. The manuscript addresses open questions concerning the initiation of convection near cold fronts, and the applied methods are suitable to address the research questions. Generally, the manuscript is well-structured and the figures are presented clearly. The research questions are clearly stated and of relevance for the readership of *Weather and Climate Dynamics*. I recommend publication of the manuscript in *Weather and Climate Dynamics* but I have several comments that should be addressed prior to publication, which are outlined in detail below.

*We thank the reviewer for taking the time to review the manuscript and for their very useful and detailed comments. We are glad that the reviewer supports the publishing of this article in WCD subject to revisions.*

**1   General comments**

1 General structure of the manuscript
Overall, the manuscript is well-structured. Yet, the Results Chapters include individual phrases and paragraphs that rather belong in the Introduction or Methods or Discussion section. I would ask the authors to make sure that the information is consistently placed in the appropriate sections and revise/streamline the Results Chapter accordingly. This does not pertain to the overall structure of the manuscript (which I think is good) but rather to several individual sentences/paragraphs. For example, Section 3.4 starts with one paragraph of introduction to Q-vector convergence which could be substantially shortened by moving the content to the Introduction and/or Methods Sections. Similarly, I would ask the authors to make sure that their discussion of results is consistently placed in the Discussion Section (and not in the Results Section).

*We thank the reviewer for their constructive feedback regarding the structure of the manuscript. Regarding Section 3.4, since large-scale lifting is*

*not only relevant for the ascending motion itself but also increasing instability, we find it relevant to briefly emphasise this aspect in the results as it is key to interpreting the results. Nevertheless, we agree that these introductions to variables can be substantially shortened, and some points can be moved to other sections. This will be changed in the revised manuscript.*

*Regarding the discussion, the discussion section serves a specific purpose in this study to frame the results in the context of Pacey et al. (2023). Some discussion regarding the findings relative to other literature (not from Pacey et al. 2023) is therefore included in the results section. We will rename the current discussion section to "Relating the results to cold-frontal cell climatology"" to highlight that this is not a discussion in the traditional sense.*

**2 Introduction of data set**

The presented study follows up on a study published by Pacey et al. (2023). I think the reader would profit from summarizing the key results from the predecessor study (Pacey et al., 2023) in more detail.

*On L56–60 the results of Pacey et al. (2023) which are relevant for this study are already summarised. The confusion may arise from our use of "For example" on L56 which we suggest removing in the revised manuscript since the results relevant for the current study are already stated.*

**3 Quantification of uncertainty**

Most figures show only mean values binned in distances from the front. I would appreciate if a measure of uncertainty (e.g., standard deviation as shading, percentiles, etc.) could be additionally shown to illustrate the associated uncertainty and variability. I appreciate that significant differences are emphasized in the figures, however, including a measure of uncertainty would be beneficial, for example, as shown by the boxplots in Fig. 5. Figure 5 also shows that the distributions are not Gaussian, i.e., it may be useful to show the median in addition to the mean for some variables that are not normally distributed.

*We thank the reviewer for the nice suggestions. Uncertainty shadings are not shown due to the overlap of distributions which would make the plots overcrowded and hard to interpret. We show an example of the 25th and 75th percentiles shown by horizontal lines for the vertical velocity below (Figure R1) as well as an example with the median shown by a triangle (Figure R2). We opt to only add the median to each plot in the main paper since this allows better interpretability and still highlights whether a distribution is non-Gaussian and whether it is positively or negatively skewed. Nevertheless, we will include discussion in the manuscript that some of the distributions overlap. Furthermore, the plots with horizontal lines for the 25th and 75th quartiles will be included in the supplementary material.*

[Figure]

*Figure R1: Vertical velocity at 850, 700 and 500 hPa depending on distance from the 700 hPa front (km) for cell grid points (orange), cell region grid points (purple) and non-cell region grid points (green). Positive and negative values indicate ascending and descending motion respectively. Horizontal lines indicate the 25th and 75th percentiles of the distributions.*

[Figure]

*Figure R2: As Figure R1 but triangles indicate the median of the distributions.*

4  Definitions of CIN (vs CAPE)

The authors apply only one (of many definitions) for CAPE but several definitions of CIN. I would ask the authors to better justify why different definitions are used for CIN (but not CAPE). I'm not sure the manuscript profits substantially from the other definitions of CIN (i.e., the authors could consider moving panels (b) and (c) of Fig. 5 to the Appendix to streamline the manuscript), as the key conclusions do not depend on the detailed definition of CIN. Instead, I would be interested to see a comparison of the CIN distribution of convective grid points with the CIN distribution of the convective environment (and non-convective

environment). Yet, the authors have mentioned that the computation of CIN for that many grid points is computationally demanding, and I understand if this is not feasible here.

*The only CAPE available on the Climate Data Store is the MUCAPE and is available for all grid points in our study thus allowing a comparison between cell grid points, cell regions and non-cell regions (Figure 4). On the other hand, the CIN on the Climate Data Store assigns a missing value if CIN exceeds 1000 J kg$^{-1}$ or where there is no cloud base. For the data points considered in our study 86% have a missing value and it is not clear how to deal with these. For this reason, we requested a different dataset (as discussed in Section 2.4.1) but only for cell grid points. Since this dataset also includes CIN considering different departure levels we decided to make use of the full dataset.*

*We thank the reviewer for the nice suggestion to move panels (b) and (c) of Fig. 5 to the appendix. We agree that this would allow greater consistency with the CAPE plot in Figure 4. We will also move Figure 10a to the appendix in the wind shear section for the same reason.*

5 Conclusions

In the conclusions, the authors summarize the most relevant factors for convective initiation by discussing each variable separately (e.g., CAPE, RH, Q-vector convergence, etc.) following the structure in the Results Section. I would find it more helpful if the authors could cluster the key factors and differences for pre-surface-frontal, near-700hPa-frontal and post-700frontal as substantial differences between these categories are presented (i.e., similar to the structure in the Discussion Section). Finally, the authors have clearly defined three research questions in the introduction, which could be addressed again in the conclusions.

*We thank the reviewer for their comments regarding the structure of the conclusions. We agree that it would be useful to refer to the research questions posed in the introduction and restructure per front relative region. This will be incorporated in the revised manuscript.*

6 Writing style

This is only a minor point, but I would ask the authors to capitalize the first letter if a specific section or chapter is referenced (e.g. Section 1, instead of section 1). Moreover, please double-check the usage of ";" throughout the text (see also specific comments below).

*Thank you for the comments on styling. According to the submission guidelines the acronym "Sect." should be used in text. We will pay careful attention to this when revising the manuscript.*

**2 Specific comments and technical corrections**

1. l. 5 "At other front relative regions": At this point in the abstract, it is unclear what "other front relative regions" refers to. Please rephrase or add additional explanations in the abstract.

   *Thank you for this comment. We propose to change this to "Behind the surface front, cells form…..". This conveys the meaning without having already read the rest of the manuscript.*

2. **Introduction**

3. l. 15: "; primarily due to convective permitting models (CPMs) at increased resolution": This does not read like a complete sentence. I would suggest to avoid the usage of ";" in the text (please also check later occurrences in the manuscript). I also think that it is common to use "convection-permitting" (instead of "convective permitting").

   *We will change to "convection-permitting" as this is indeed more common and change the semi-colon to a comma for increased readability. Thank you for the suggestions.*

4. l. 28: Please remove "etc" and specify.

   *Etc in this case indicates there are several other quantities that could be listed. However, since "such as" is already written before the list, the "etc" can be removed. Thank you for the suggestion.*

5. l. 84: Please avoid double-brackets ") (". See also l. 129, 258, 357.

   *We appreciate the suggestion, however there is no mention of double brackets in the submission guidelines so we will leave it up to the typesetters to decide on this styling aspect.*

   *https://www.weather-climate-dynamics.net/submission.html*

6. l. 32 ff: This reads a bit colloquially, please rephrase.

   *We suggest revising to "Anticipating the spatiotemporal onset of convective cells is essential for…."*

7. l. 43 f: "The literature would benefit from studies quantifying the relevance of frontal lifting at different regions relative to the front, especially during the warm-season.": This sentence appears a bit out of context, I'm not sure if it necessary here.

   *We will move this sentence to the end of the paragraph. Thank you for picking up on this.*

8. l. 51 ff: The role of wind shear should be discussed together with other factors relevant for convection, i.e., in the paragraph starting in l. 27. Moreover,

following General Comment 1, I would ask the authors to include the background information provided in the Results sections in the introduction. Overall, the introduction would benefit from more clearly summarizing and structuring key variables relevant for convection, targeting specifically convection embedded in the frontal environment.

*We will reorganise and streamline the introduction section as well as incorporate additional literature. Thank you for the suggestion.*

9. **Methods**

10. l. 85: I would ask the authors to include one explanatory sentence on the TFP equation.

    *We will add that "The term represents the rate of change of τ projected in the direction of the thermal gradient," Thank you for the suggestion.*

11. l. 106: "The process is repeated 30 times to remove any local-scale features.": Please explain here, why specifically "30 times" was chosen. I would ask the authors to elaborate on why this specific smoothing method was applied (in contrast to other methods)?

    *This was chosen subjectively based on looking at several case studies. There is no standard practice when smoothing as it depends on the resolution of the dataset. More smoothing reduces the strength of gradients. Smoothing more times while using a lower gradient threshold would yield similar results. Smoothing becomes particularly important when using convection-permitting models with higher resolution. We will note that this choice was arbitrary. Thank you for raising this point.*

12. l. 109: "than some previous studies": Please include those studies here.

    *We will provide Schemm et al. (2016) as an example who tested 300 km and 500 km front length criteria. Thank you for the suggestion.*

13. l. 118 f: "This is also supported by the mean maximum climatological surface convergence in ERA5 data (Pacey et al., 2023; their Figure 3)". Please add some additional information on how this figure supports the statement, such that the reader does not have to read the mentioned publication themselves.

    *The reader does not necessarily need to read the publication unless they would like to see the visualisation of this result. We will make this statement more explicit by saying "The mean maximum climatological surface convergence in ERA5 was found 300 km ahead of the 700 hPa front in Pacey et al. (2023) (their Figure 3), which supports the aforementioned assumption."*

14. l. 132 ff. Please label the criteria following your approach above (i.e., "(A)", "(B)", etc.), and add text describing the criteria.

*Thank you for the suggestion, we will incorporate this in the revised manuscript.*

15. l. 142: Does the time of first detection correspond to the first exceedance of 46 dBZ? Would it be possible to track these cells also before they reach maturity, i.e. in their developing phase with lower reflectivity?

    *Yes, it is the first exceedance. In principal, a lower threshold could be used to start tracking cells in their developing phase, but this information is not incorporated into the KONRAD2D dataset.*

    l. 139 ff: "Since some cells have a lower area than the grid size the bounds are increased by 0.125 degrees (half a grid point)": I find this difficult to understand, please rephrase.

    *We suggest to rephrase this to "To take the different spatial resolutions of the datasets into account and the fact that some cells are smaller than the ERA5 grid size, the cell boundaries are extended by 0.125 degrees in each direction"*

16. Figure 1: This figure nicely illustrates the definition of the three categories. For further illustration, the authors could additionally include the position of the surface cold front as well as the defined pre-frontal, pre-surface frontal and post-frontal regions. *Thank you for the positive feedback regarding Figure 1. We will add a surface frontal line to Fig. 1a and if there is space it would also be nice to add pre-surface-frontal, pre-700-frontal post-700-frontal text labels. Thank you for the suggestion.*

17. l. 152, Table 2: I appreciate that detailed numbers of grid points in each category are provided in Table 2, yet it would be easier to comprehend if the numbers were shown as an additional figure.

    *Thank you for the nice suggestion. This would indeed be more informative and match the style of other figures in the manuscript.*

18. l. 161 ff: The choice of variables should clearly be motivated in the introduction, such that refering to literature in this short paragraph is not required anymore.

    *See our response to general comment 1. We argue it is important to briefly introduce each variable, however we will make sure to substantially shorten such introductions. Thank you for the suggestion.*

19. l. 166: As mentioned above, it is not fully clear to me why different definitions of CIN, but not of CAPE are used.

    *This is primarily due to dataset availability, see our response to general comment 4 for further details.*

20. l. 168: "So that a CIN value is present for all grid points": Please rephrase (e.g., "To ensure that ...").

    *Thank you for the suggestion, we will revise the updated manuscript.*

21. Equation 3: Please define the symbols/abbreviations in the text.

   *We appreciate symbols are typically defined in this way. However, since there is a rather long list of symbols to be defined, showing them as a column list increases readability. We thus remain with the original version.*

22. l. 193 f. I would ask the authors to elaborate on the smoothing method and how the number of smoothing cycles was exactly determined (see also previous comment above)?

   *This is again subjective as it depends on the dataset resolution. In this case, we have already elaborated and noted that smoothing values between 10 and 100 were tested. We will add that the decision is ultimately subjective and that the same smoothing filter as used in section 2.1 is used. Thank you for this suggestion.*

   **Results**

23. l. 203: It could be helpful for the reader to show and discuss the occurrence frequency of convection (Fig. 12) before Sub-sections 3.1, 3.2, etc. to familiarize the reader with the dataset.

   *We thank the reviewer for the suggestion but since the discussion section, which will be renamed, serves the purpose of putting the results in the context of Pacey et al. (2023) we see it more fitting to include it in that section.*

24. l. 207 ff: The authors emphasize the differences in dewpoint between convective and non-convective environments. Did the authors also consider the 2-m temperature distributions? Are the differences in dewpoint related to differences in humidity (i.e., dew point depression) or to differences in the background temperature? I would be interested to see Fig. 2 for 2-m temperature.

   *We show the surface temperature with the 25$^{th}$ and 75$^{th}$ quartiles shown as horizontal lines (Figure R3). A similar result is observed to the surface dewpoints with a significant positively anomaly at cell grid points at all front relative regions. The magnitude of the anomaly is very similar as well (around 3–4 $^{o}$C). We will mention this result in the dew point section and include Figure R3 (except with median triangles) in the appendix.*

[Figure]

*Figure R3: As Figure R1 but for surface air temperature.*

25. l. 236-241. This is repetitive. Please streamline and/or move to Methods and Discussion Sections.

    *We agree some of this can be shortened, however as mentioned in response to a previous comment we think it is useful to briefly introduce the variables in a few sentences. Thank you for the suggestion.*

    l. 241 f: Please rephrase and avoid using ";".

    *We will replace the semi-colon with a comma here. Thank you for the suggestion.*

26. l. 259: I assume the two numbers have been swapped? I guess post-700frontal should have lower absolute CIN values.

    *Thank you for picking up on this, we will amend this.*

27. l. 262: "by forecasters" is not necessarily required.

    *This is just to emphasise its usage in operational forecasting.*

28. l. 263-271: Please streamline this paragraph.

    *As mentioned in previous comments, we will do our best to streamline this paragraph and remove lengthy descriptions in the revised manuscript.*

29. l. 279: "shift to the left on the plots": Please rephrase.

    *We will rephrase this to "shift towards the left (cold side) with increasing height from 850 to 500 hPa."*

30. l. 297 f: The authors show mean vertical velocity for cell grid points of on average a few cm/s. I would ask the authors to relate those numbers (in the Discussion) to typically observed convective updraft velocities and discuss its implications for using ERA5 vertical velocity for studying convection. Is vertical velocity normally distributed and are large outliers present?

    *We thank the reviewer for the suggestion. The vertical velocities in ERA5 at convective cell grid points in our study are up to 2 orders of magnitude lower than what can be seen in both observations and numerical simulations of convective updrafts (e.g. Weisman and Klemp, 1982). Therefore, we are arguing there is some signal of partially resolving convection occurrence but not actual updrafts. We agree some brief discussion of this aspect could be useful and this will be incorporated in the revised manuscript.*

31. Caption Fig. 7: Please correct the typo: "Postive"

    *Thank you for picking up on this, we will amend this.*

32. l. 335-336: This sentence is not fully clear to me, please rephrase.

    *We believe separating this into two sentences will increase the clarity. Thank for you the suggestion.*

33. l. 351: I would appreciate if the authors could include a more original reference (in addition to EUMeTrain).

    *We will add a reference to Browning (1990). Thank you for the suggestion.*

34. l. 352: Please rephrase "condensation" by "cloud formation" or similar, as condensation inherently implies that precipitation only forms from warmphase cloud processes.

    *We will amend this, thank you for the suggestion.*

35. l. 354 ff: Fig. A2 suggests that a substantial number of identified cells occurs in an environment with large cloud cover, and thus, may be embedded in a larger precipitating cloud system (see Fig. 11), such as the warm conveyor belt (which has also been mentioned). The Discussion Section could profit from including studies on precipitation characteristics and distribution in the warm conveyor belt airstream compared to pre-frontal convection.

    *Indeed, we believe that the majority of cells between the 700 hPa front and surface front are embedded in stratiform precipitation regions. We will include additional discussion on this in the revised manuscript and cite previous literature on such topics (e.g. Oertel et al. 2020). Thank you for the suggestion.*

36. l. 361-364: I'm not sure if I agree with the conclusions in this paragraph. Assuming ERA5 would (at least partially) represent convection, the precipitation signal would show up in "Large-scale precipitation", and not in the parameterized convective precipitation. In this case, I would expect a difference between the three categories in Fig. 11a (which is very small, in particular pre-frontal). Instead, the parameterized "convective precipitation"

differs between categories (Fig. 11b), suggesting that convection cannot explicitly be represented in ERA5 (and needs to be parameterized). Apologies, in case I mis-understood this paragraph. I would ask the authors to rephrase this paragraph.

*I think this is indeed a misunderstanding. We don't argue that ERA5 can explicitly represent convection without parameterizations. Of course, convective precipitation can only appear if convection is triggered in the parameterization scheme. The triggered parameterised convection may feedback on the vertical velocity field due to condensation and latent heat release (and hence further ascent). Even with convection parameterizations though, there is no guarantee convection will be triggered in the correct place and time. We find that there is a significantly higher convective precipitation total where convective cells were observed compared to non-cell regions.*

*We will rephrase this paragraph mentioning some of the points above. Thank you for bringing this to our attention.*

l. 366 ff: Please streamline this paragraph, this information has been repeated several times.

*We suggest to explicitly state the purpose of this section which is to put the results in the context of Pacey et al. (2023). We will remove text focused on what was shown in earlier sections. Thank you for the suggestion.*

37. l. 371 ff: I appreciate that the authors discuss and summarize the relevant factors for (i) pre-surface-frontal cells, (ii) near-700hPa-frontal cells, and (iii) post-700-frontal cells. While the overall structure is good, this section could profit from (even more thoroughly) comparing the presented results to previous studies, and e.g., pick up literature that has been mentioned in the introduction.

*We thank the reviewer for this suggestion. We will further review previous literature and add further citations where required.*

38. l. 408: typo: include "$\circ$" in "16 C".

*Thank you for noticing this, this will be changed.*

39. **Conclusions**

40. l. 401: Please remove "etc" and specify.

*As the previous comment, this is used to indicate that there are several other examples which are not listed. As a previous comment, we suggest writing "such as" before the list and remove the "etc" at the end.*

41. l. 408-426: In general, I think it is ok to use bullet points to emphasize key conclusions. Yet, I would reduce the number of bullet points and more strongly aggregate the relevant information.

*We agree that some bullet points could be aggregated thus reducing the overall number of bullet points. Thank you for the suggestion.*

42. l. 422 f: It is expected that convective cells are associated with positive vertical velocity. In Section 3.5, the authors have briefly mentioned the implication for consistently seeing this signal in ERA5 data. I think the manuscript would profit from a more thorough discussion on ERA5 and its ability to (at least partially) represent convection and convective precipitation. The last sentence of the manuscript brings up this open question, yet it could be discussed more thoroughly.

    *To robustly assess how well ERA5 represents convective precipitation amounts would require observations of precipitation amounts. Here, we only use radar reflectivity which has additional uncertainties when converting it to a precipitation amount. Thus, based on our results we can only briefly speculate where this signal comes from. Therefore, we open this topic (and further discussion) for future work. As mentioned in our response to comment 36, we will further elaborate on what the convective parameterization scheme could mean for the vertical velocity field.*

43. Have the authors considered including a schematic that illustrates and summarizes the relevant factors that discriminate between convective and non-convective conditions in different regions of the front?

    *We thank the reviewer for this nice suggestion. We will work on designing such a graphic which could also serve as the highlight figure for the paper.*

*References*

*Browning, K, A,. 1990: Organization of clouds and precipitation in extratropical cyclones; in: Extratropical Cyclones, The Erik Palmen Memorial Volume, Ed. Chester Newton and Eero O Holopainen, p. 129 – 153*

*Oertel, A., Boettcher, M., Joos, H., Sprenger, M. and Wernli, H., 2020. Potential vorticity structure of embedded convection in a warm conveyor belt and its relevance for large-scale dynamics. Weather and Climate Dynamics, 1(1), pp.127-153.*

*Pacey, G., Pfahl, S., Schielicke, L., and Wapler, K.: The climatology and nature of warm-season convective cells in cold-frontal environments over Germany, Natural Hazards and Earth System Sciences, 23, 3703–3721, https://doi.org/10.5194/nhess-23-3703-2023, 2023.*

*Schemm, S., Nisi, L., Martinov, A., Leuenberger, D., and Martius, O.: On the link between cold fronts and hail in Switzerland, Atmos. Sci. Lett., 17, 315–325, 2016*

*Weisman, M. L., and J. B. Klemp, 1982: The Dependence of Numerically Simulated Convective Storms on Vertical Wind Shear and Buoyancy. Mon. Wea. Rev., **110**, 504–520.*

---

## Author Comment (AC2)

*Response to RC2*

*Authors' responses are in red italics*

**Recommendation:** minor revisions

**General**

By using automated front detection and convective cell tracking data, this study analyzes the environmental conditions and lifting mechanisms influencing convection near cold fronts in warm seasons in Germany. Results show that pre-surface-frontal cells tend to form in areas with the highest surface dew points and CAPE. Other front relative regions also support cell formation, though with lower CAPE and dew points compared to non-cell regions. Mid-level humidity helps distinguish post-frontal cell locations from non-cell regions. Pre-surface-frontal cells experience strong large-scale lifting at 850 hPa and 700 hPa, with high convective inhibition. Significant large-scale lifting also appears post-frontal, especially at 500 hPa. Additionally, less sunshine was observed before cell initiation compared to non-cell regions, suggesting that solar heating may not drive most cold-frontal cell initiation.

This study provides a valuable analysis of the atmospheric conditions that influence convection near cold fronts. I find the subject interesting and well within the scope of WCD. It is well written, has a clear structure, and good illustrations. However, I have a couple of minor concerns which can maybe be solved with more explanations. I expand on some of these concerns below and outline additional minor/technical comments.

*We thank the reviewer for taking the time to review the manuscript and for their construct comments and feedback. We are glad that the reviewer supports the publishing of this article in WCD subject to revisions.*

**Specific comments**

- The authors study the warm-season in Germany, but only convection related to frontal zones. I assume that a large part of convection occurs in situations with weak synoptic forcing as well. Can the authors comment on the overall relevance of frontal convection in Germany? Moreover, several field experiments were carried out to better understand convection initiation in Germany. At least some of those with their main findings should be cited.

  *The primary focus of the current study is on cold-frontal cell environments and lifting mechanisms. However, in Pacey et al. (2023) we did look at climatological differences between cold-frontal and non-cold-frontal convective cells. For example, we found on days where cells initiated in proximity to cold fronts over twice as many cells are detected compared to non-cold-frontal cell days. Furthermore, we found differences in the fraction of cell days associated with cold fronts depending on the location in Germany, with cell days in north-western Germany and southern Germany being most and least associated with cold fronts, respectively. We will include some brief discussion of these findings in the introduction.*

*Regarding field campaigns in Germany (e.g. COPS; Wulfmeyer et al. 2011), we have focused the literature in the introduction on previous studies on the environments in which convective storms form over longer timeseries since this is a primary focus of this study and not on individual case studies and field campaigns. However, we agree that more background on the general topic of convective initiation in Germany would be useful, particularly in the 3$^{rd}$ paragraph of the introduction. In the revised manuscript, we will include discussion of COPS regarding frontal lifting and local lifting from orography. Thank you for the nice suggestion.*

*We note that it would also be an interesting focus for future work to look at non-cold-frontal cell environments and lifting mechanisms in comparison to cold-frontal.*

- Some general information about convection initiation is given in the introduction. However, the multiple effects of mountains or land-surface heterogeneities are not mentioned. I recommend enlarging that section with these points, particularly the role of low-level convergence zones.

*Like the point above, we will include discussion of low-level convergence zones that can be created near orography (e.g. Figure 1; Wulfmeyer et al. 2011). Thank you for the nice suggestion.*

- One aspect I may have missed in the manuscript is the fact that even if CAPE values are lower in post-frontal regions, the atmosphere is often unstable due to the advection of colder air at higher levels. Warming by solar radiation is then often sufficient to initiate convection in large areas.

*Indeed, the cold air advection aloft combined with surface heating can increase CAPE. The insolation can also act as a trigger for convection. We would assume this is particularly important for increasing CAPE on the post-700-frontal side, whereas moisture advection would be more important on the pre-700-frontal side. The origin of CAPE generation is not a key focus of this study, so we have not addressed it in too much detail. We do however mention the importance of solar heating on the post-700-frontal side in section 4.3.*

- I am a bit concerned about the fact that pre-surface-frontal cells form in environments with the highest CAPE AND strongest CIN. High CAPE values usually occur at low altitudes of the LFC which also often is associated with smaller values of CIN. Please comment on this.

*We are not sure if we have fully understood the reviewer's comment. Maybe the reviewer is suggesting a lower LFC may allow higher CAPE as in theory the vertical distance over which a parcel is positively buoyant could be larger. CAPE is dependent on different factors though, e.g. surface heating, near-surface moisture and lapse rates (particularly in the mid-levels). If strong CIN is present, this means more lifting is required to lift parcels to their LFC where they can then utilise the CAPE. In turn, this could mean that convection is not triggered until later in the day following further solar heating and moisture advection (thus increasing CAPE). Figure 2 (surface dew points) seems to indicate that the surface moisture availability is one key contributor higher CAPE. We hypothesise that the stronger CIN for*

*example could allow additional moisture build up near the surface before convection is triggered.*

- The authors detect and track convective cells based on radar data from the DWD. Radar-derived precipitation adjusted to surface observations is also available from the DWD on the same domain. I wonder why this data set is not used for precipitation? Precipitation and in particular convective precipitation is certainly not best represented in the ERA5 data set.

*We assume the author is specifically referring to section 3.8 and Figure 11. As the reviewer mentions, convective precipitation is not expected to be represented well in ERA5. The aim of section 3.8 is to assess how it is represented at cell grid points, cell regions and non-cell regions. If ERA5 exhibited no skill at all, we would expect no significant difference in the mean between convective grid points and non-cell regions. The purpose of 3.8 is not to analyse the exact rainfall totals, for which a combined radar/rain gauge product would be more suitable as the reviewer suggests. We suggest adding an extra sentence in section 3.8 to make this clearer. Thank you for mentioning this point.*

- Why do you need a smoothing of $\vartheta_e$ and why 30 times?

*This was chosen subjectively based on looking at several case studies. There is no standard practice when smoothing as it depends on the resolution of the dataset. More smoothing reduces the strength of gradients. Smoothing 50 times while using a lower gradient threshold would have yielded the similar results. Smoothing becomes particularly important when using convection-permitting models with higher resolution. We will note that this choice was subjective. Thank you for raising this point.*

- P6: You state that cold fronts reach the southern parts of Germany less frequently. Is this related to the more complex terrain there?

*Indeed, we believe this is related to the terrain. In Pacey et al. (2023), we clustered different front types and found a common type where fronts become distorted and curve around the shape of the Alps. The reviewer is referred to Figure 9 of Pacey et al. (2023) if they are still interested.*

- P6, L162: You state that wind shear affects convective initiation. Wind shear, however, is not a trigger mechanism, it is decisive for the evolution and organisation of the initiated convection. I suggest to write that it affects convection and not convective initiation.

*There has been some research showing that wind shear is relevant for the transition from shallow to deep convection, we reference a good overview of the topic on L163 (Peters et al. 2022). As an example, one way shear may negatively contribute towards this transition is through increased entrainment in high shear environments (e.g. Markowski and Richardson, 2010; their section 7.2.1). Of course, this also depends on the background environmental relative humidity. Further research is still required in this area, but this is not a key focus of our study. Nevertheless, we still think it is important to highlight that the relevance of wind shear extends beyond convective organisation.*

- As for convective precipitation, I doubt that the vertical velocity in the ERA5 data set is really representative for deep convection. Please comment.

*There may have been a misunderstanding regarding our remarks on the representation of convection in ERA5. We would not expect precipitation rates in ERA5 to be equivalent to those that would be observed where convective cells were detected. Likewise, the vertical velocity would not be comparable to individual updrafts. Nevertheless, given there is a significant difference in the convective precipitation mean between convective cell grid points and non-cell grid points shows some signal of convection being triggered in the parameterization scheme in the right place and time. The triggered parameterised convection may then feedback on the vertical velocity field due to condensation and latent heat release (and hence further ascent).*

*We will rephrase this paragraph mentioning some of the points above. Thank you for bringing this to our attention.*

- What are the implications for forecasting convective storms near frontal zones?

Are there any ways to improve numerical models with these findings?

*We thank the reviewer for posing these interesting questions. The findings in study highlight the complexity of the environments in which convective storms form since they vary depending on the distance from the front. For example, we find greater importance of upper-level large-scale lifting post-frontal compared to pre-surface-frontal. A forecaster therefore could focus more on upper-level lifting when assessing the probability of convective cell initiation post-frontal.*

*This reality complicates representation of convection in numerical models since parameterizations should (in theory) be applicable globally. Here, we show even in Germany the importance of different factors controlling cell initiation varies depending on the front relative region.*

**Technical comments**

- P3, L85: Paramter → Parameter

*Thank you for noticing this typo, this will be revised.*

- Line breaks occur between numbers and their units troughout the entire manuscript (e.g. P4, L117-118). Please correct that everywhere.

*Thank you for picking up on this, we will revise this.*

- P6, L151: ...bin at  the current...

*Thank you for noticing this typo, this will be revised.*

- P6, L165: A full list...  **is** shown...

*Thank you for noticing this typo, this will be revised.*

- P6, L177: The quasi-geostrophic forcing for ascending and descending motion can be **measured** using the Q-vector convergence... I think "measured" is not the best word here as this is not a measurement. Maybe "expressed" or "described" are better options.

    *We agree "expressed" is a better word here. Thank you for the suggestion.*

- P10, L286: This result highlights the importance of upper-level forcing **particularly** on the development of convective cells **particularly** at the 700 hPa front and also post-700-frontal. Please rephrase.

    *We will remove both occurrences of 'particularly' from the text since it is not necessary to convey the point. Thank you for pointing this out.*

- P14, L408: 16 C → 16°C

    *Thank you for pointing this out. This will be amended.*

- P17, Fig. 2 caption: celcius → Celsius

    *Thank you for noticing this typo, this will be revised.*

    • P20, Fig. 7 caption: Postive → Positive

    *Thank you for noticing this typo, this will be revised.*

    • P29, L509: 1.  edn.

    *Thank you for noticing this typo, this will be revised.*

*References*

*Pacey, G., Pfahl, S., Schielicke, L., and Wapler, K.: The climatology and nature of warm-season convective cells in cold-frontal environments over Germany, Natural Hazards and Earth System Sciences, 23, 3703–3721, https://doi.org/10.5194/nhess-23-3703-2023, 2023.*

*Peters, J. M., H. Morrison, T. C. Nelson, J. N. Marquis, J. P. Mulholland, and C. J. Nowotarski, 2022: The Influence of Shear on Deep Convection Initiation. Part I: Theory. J. Atmos. Sci., **79**, 1669–1690*

*Wulfmeyer, V., Behrendt, A., Kottmeier, C., Corsmeier, U., Barthlott, C., Craig, G.C., Hagen, M., Althausen, D., Aoshima, F., Arpagaus, M., Bauer, H.-S., Bennett, L., Blyth, A., Brandau, C., Champollion, C., Crewell, S., Dick, G., Di Girolamo, P., Dorninger, M., Dufournet, Y., Eigenmann, R., Engelmann, R., Flamant, C., Foken, T., Gorgas, T., Grzeschik, M., Handwerker, J., Hauck, C., Höller, H., Junkermann, W., Kalthoff, N., Kiemle, C., Klink, S., König, M., Krauss, L., Long, C.N., Madonna, F., Mobbs, S., Neininger, B., Pal, S., Peters, G., Pigeon, G., Richard, E., Rotach, M.W., Russchenberg, H., Schwitalla, T., Smith, V., Steinacker, R., Trentmann, J., Turner, D.D., van Baelen, J., Vogt, S., Volkert, H., Weckwerth, T., Wernli, H., Wieser, A. and Wirth, M. (2011), The Convective and Orographically-induced Precipitation Study (COPS):*

the scientific strategy, the field phase, and research highlights. Q.J.R. Meteorol. Soc., 137: 3-30. https://doi.org/10.1002/qj.752